# Characteristics and Differences Analysis for Thermal Evolution of Wufeng–Longmaxi Shale, Southern Sichuan Basin, SW China

**Peixi Lyu [1,2], Jianghui Meng [1,2,]\*, Renfang Pan [1,2], Xuefei Yi [1,2], Tao Yue [1,2] and Ning Zhang [1,2]**

[1] Hubei Cooperative Innovation Center of Unconventional Oil and Gas, Yangtze University, Wuhan 430100, China; lpxyzu@163.com (P.L.); pan@yangtzeu.edu.cn (R.P.); xuefeiyi1986@163.com (X.Y.); ytlishp@163.com (T.Y.); zhangnyzu@163.com (N.Z.)

[2] Ministry of Education Key Laboratory of Oil and Gas Resources and Exploration Technology, Yangtze University, Wuhan 430100, China

\* Correspondence: mjh@yangtzeu.edu.cn

**Abstract:** The marine shale of the Upper Ordovician Wufeng Formation–Lower Silurian Longmaxi Formation is the main source rock and the target of shale gas exploration in the southern Sichuan Basin. The maturity of organic matter (OM) is a vital indicator for source rock evaluation. Due to the lack of vitrinite, the organic matter maturity of the Wufeng–Longmaxi Formations in the southern Sichuan Basin is difficult to accurately evaluate. In total, 33 core samples of the Wufeng–Longmaxi Formations in the southern Sichuan Basin were selected to observe the optical characteristics of solid bitumen and graptolites and measure their random reflectance. Simultaneously, Raman spectroscopic parameters of kerogen were also used to quantitatively analyze the change in maturity. By using Raman spectroscopic parameters as mediators, conversion equations between graptolite random reflectance ($GR_{or}$) and equivalent vitrinite reflectance ($EqVR_o$) were established. Taking the calculation results of $EqVR_o$ as constraints, the tectono-thermal evolution history of Wufeng–Longmaxi Shale in the southern Sichuan Basin is constructed through basin modelling. The results show that the maturity of Wufeng–Longmaxi Shale in the western Changning, Luzhou-western Chongqing, eastern Changning and Weiyuan areas decreases successively. The $EqVR_o$ falls in the ranges of 3.61%~3.91%, 2.92%~3.57%, 3.08%~3.25%, 2.41%~3.12%, and the average $EqVR_o$ is 3.73%, 3.30%, 3.18% and 2.80%, respectively. Thermal evolution in western Changning was controlled by the thermal effect of the Emeishan mantle plume and paleo-burial depth, while the thermal evolution of other areas was mainly controlled by paleo-burial depth. This study provides a reliable parameter for the evaluation of thermal maturity and makes a more accurate calibration of the maturity of the Wufeng–Longmaxi Formations in the southern Sichuan Basin; it also expounds the factors for the differences in thermal evolution in different parts of the area.

**Keywords:** graptolite; solid bitumen; Raman spectroscopy; basin modelling; thermal maturity; Wufeng–Longmaxi Formations; Sichuan Basin



## 1. Introduction

The black graptolite-bearing and organic-rich shales of the Upper Ordovician Wufeng Formation ($O_3w$)–Lower Silurian Longmaxi Formation ($S_1l$) are the most important hydrocarbon source rocks and shale gas exploration and development targets in China [1,2]. In the Sichuan Basin, the Wufeng–Longmaxi Formations are extensively deposited [3,4]. The evaluation of the maturity of OM is crucial for the hydrocarbon generation potential of source rocks [5,6]. In addition, the increase in thermal maturity promotes the formation and evolution of organic pores in shales [7,8], while the amount of adsorbed gas at first increases, and then decreases [5,9]. The determination of black shale maturity is the basis for a dynamic study of the process of hydrocarbon generation, discharge, retention and migration, and it is of great significance for shale gas exploration and evaluation.

Vitrinite reflectance ($R_o$) is a very commonly used indicator of thermal maturity [10,11]. However, due to the lack of vitrinite in pre-Devonian rocks, $R_o$ cannot be measured directly. Therefore, for a long time, it was difficult to accurately evaluate the maturity of the Wufeng–Longmaxi Formations. Moreover, the Wufeng–Longmaxi Formations in the southern Sichuan Basin are commonly in the over-mature stage, which makes the applicability and accuracy of other maturity indicators, such as the conodont color-alteration index [12], Rock-Eval $T_{max}$ [13,14], fluorescence photometry [15], and biomarker parameters [16], unsatisfactory in this area.

At present, the maturity levels of the Wufeng–Longmaxi Formations are typically assessed by $EqVR_o$ calculated from solid bitumen reflectance ($BR_o$) [17–19], although the results from different researchers vary largely. One reason might be that the source and origin of solid bitumen are complex [17,20–22]; multi-source and multi-stage solid bitumen usually coexist, and the occurrence forms of solid bitumen are different in different lithology and different maturity stages. Another reason is that solid bitumen develops obvious optical anisotropy and heterogeneity if its reflectance exceeds 1.5%, due to the solid bitumen porosity developing as a result of secondary cracking and the generation and expulsion of hydrocarbons [21–23]. In the southern Sichuan Basin, the Wufeng–Longmaxi Formations are in an over-mature stage; the solid bitumen has small particles and strong heterogeneity. The uncertainty of $BR_o$ gives rise to a lack of uniform understanding and systematic research on the thermal maturity of the Wufeng–Longmaxi Formations in this area.

Graptolite is one of the most common bioclasts in the Wufeng–Longmaxi Formations. Due to its high abundance, obvious characteristics, wide distribution and comparability, graptolite was widely used by scholars at home and abroad for biostratigraphic correlation, paleoecological judgment and sedimentary environment evolution studies [24–26]. With the exploration of, and developments in, Lower Paleozoic shale gas in recent years, an increasing number of scholars re-focused on the significance of graptolite reflectance as a maturity indicator [20,27–29]. However, there is no clear and unified view on the relationship between graptolite reflectance and $EqVR_o$ and its scope for application in the high–over-mature stage. The maturation paths of graptolite and vitrinite are different in high-over maturity stage: graphitization in graptolite occurs much later than that in vitrinite, which causes the graptolite reflectance and $R_o$ to, potentially, not follow a single linear relationship [28,30,31]. The existing conversion equations between $GR_o$ and $EqVR_o$ from different researchers [20,27,32–34] may not be sufficiently accurate.

Raman spectroscopy attracted increasing attention in the field of thermal maturity evaluation of OM due to its advantages; specifically, its simple preparation and tiny measuring point, as well as the fact that it does not destroy samples and is a convenient and fast method of testing [35–39]. Raman spectroscopic parameters are highly accurate in the high–over-mature stage, so they are particularly suitable for calculating the reflectance of the high–over thermal evolution shales [35]. Previous studies usually required the differentiation of organic macerals [36,40,41]. However, it is difficult to accurately distinguish the OM particles in high–over thermal evolution shales due to their small size and similar optical properties. Liu et al. [35] and Wang et al. [42] carried out Raman spectroscopy experiments on different macerals in shales at the high–over-mature stage, but no significant difference was found in the results of different macerals. The consistency of results confirms that for the high–over-mature samples, Raman spectroscopy performed on kerogens can provide rapid estimates of equivalent vitrinite reflectance, and there is no need to distinguish different organic macerals.

In conclusion, there are limitations to using a single indicator to study the thermal maturity and thermal evolution of the Wufeng–Longmaxi Formations. In order to truly evaluate the maturity of the Wufeng–Longmaxi Formations in the southern Sichuan Basin, this paper presents a systematic analysis of the reflectance of graptolites and solid bitumen, as well as Raman spectroscopy characteristics of kerogens. Taking Raman spectral parameters of Raman band separation and ratios as mediators, we establish the correlations

between $GR_{or}$ and $EqVR_o$ for over-mature shales. Through the correlations, we make a more objective and accurate evaluation of the thermal maturity of the Wufeng–Longmaxi Formations in the southern Sichuan Basin. On this basis, we reconstruct the burial history, thermal evolution history and hydrocarbon generation history of the Wufeng–Longmaxi Formations in different areas of the southern Sichuan Basin through basin modelling.

## 2. Geological Settings

The Sichuan Basin, comprising an area of approximately $18 \times 10^4$ m$^3$ in southwest China, is a superimposed basin developed on the basis of the Upper Yangtze Craton [43]. The Basin is tectonically bound by the Emeishan–Liangshan Fold Belt in the southwest, the Longmenshan Fold Belt in the northwest, the Micangshan Uplift in the north, the Dabashan Fold Belt in the northeast and the Hubei–Hunan–Guizhou Fold Belt in the southeast. Since the Proterozoic, the Sichuan Basin experienced three main tectonic evolution stages: a marine carbonate platform in an extensional setting from the Ediacaran to the Middle Triassic, a foreland basin with fold and thrust deformation from the Late Triassic to the Late Cretaceous, and subsequent exhumation and structural modification from the Late Cretaceous to Quaternary. Therefore, the Wufeng–Longmaxi Shale experienced substantial deep burial during the Early Mesozoic and an intense uplift and erosion from the Late Mesozoic to the Cenozoic [44]. The current burial depths of the Wufeng-Longmaxi Shale are within a range of 1300–4500 m in the Sichuan Basin [45], and the shale is buried at a shallow depth and even occurs as outcrop in strongly uplifted areas.

In the late Early Ordovician–Silurian, due to the plate convergence between the Huaxia plot and the Yangtze plot, the Upper Yangtze Craton was in a tectonic setting of compressional uplift. The Xuefeng uplift, Chuanzhong uplift and Qianzhong uplift were exposed over the sea, making the broad sea features during the Middle Ordovician transform into restricted shallow sea surrounded by the paleo uplift, forming a large sedimentary environment of low energy, under-compensation and anoxia [46]. Affected by this tectonic movement and transgression, black graptolite-bearing shale was deposited during the Late Ordovician as the Wufeng Formation [4], its thickness ranging from a few to tens of meters. The lower part of the Wufeng Formation is black shale, interbedded with thin layers of volcanic ash deposits. The upper part of the Wufeng Formation is limestone or marl. The Guanyinqiao Member is at the top of the Wufeng Formation; its lithology is shell-rich marlstone and calcareous mudstone with a thickness of tens of centimeters, and the Hirnantia fauna is rich in fossils. The Longmaxi Formation was deposited during rapid marine transgression following the Late Ordovician glacial deposits of the Guanyinqiao Member, as concluded from its upward coarsening and prograding sequence. From the bottom up, the Longmaxi Formation is divided into Long1 and Long2 members. The Long1 member mainly comprises organic rich black carbonaceous shale, siliceous shale and dark gray arenaceous shale containing abundant graptolites and has a thickness of 30–120 m [47]. The Long2 member mainly comprises grayish-yellowish green shale and arenaceous shale interbedded with siltstone and marlstone. The sand content in the Long2 member increases from the bottom to the top, forming an upward coarsening sedimentary sequence. The Long1 member is subdivided into Long1$_1$ and Long1$_2$ sub-members, and the Long1$_1$ sub-member is further subdivided into Long1$_1$$^1$, Long1$_1$$^2$, Long1$_1$$^3$, and Long1$_1$$^4$ beds [48]. According to statistical data, TOC values of the Wufeng–Longmaxi Shale range from 0.4% to 18.4%, with an average of 2.5%. The high TOC sections (over 2%) developed mainly during the lower part of the Lower Silurian Longmaxi Formation. The TOC content gradually decreases upward with the increase in the calcareous and silt content [1,49].

The southern Sichuan Basin is located in the southwest margin of the Upper Yangtze Platform in southwest China (Figure 1), to the east of the Daliang Mountains, south of the denudation line of the Longmaxi Formation in the Leshan–Longnüsi paleo-uplift, west of Huaying Mountain, and north of the Qianbei sag. The area is about $4 \times 10^4$ km$^2$, located in the southern Sichuan low-steep structural belt with small-scale fractures. It was tectonically uplifted in a late stage and, hence, there is an extensive tectonically quiet region with a small

magnitude of uplift [50,51]. The study area is divided into four parts: Luzhou, western Chongqing, Changning and Weiyuan. The main research sections are the upper Wufeng Formation and $Long1_1{}^1$–$Long1_1{}^4$ beds, with a thickness of 40–130 m and TOC value over 2%. The current burial depth of the Wufeng–Longmaxi Shale is generally less than 3500 m in Weiyuan and Changning, and more than 3500 m in Luzhou and western Chongqing.

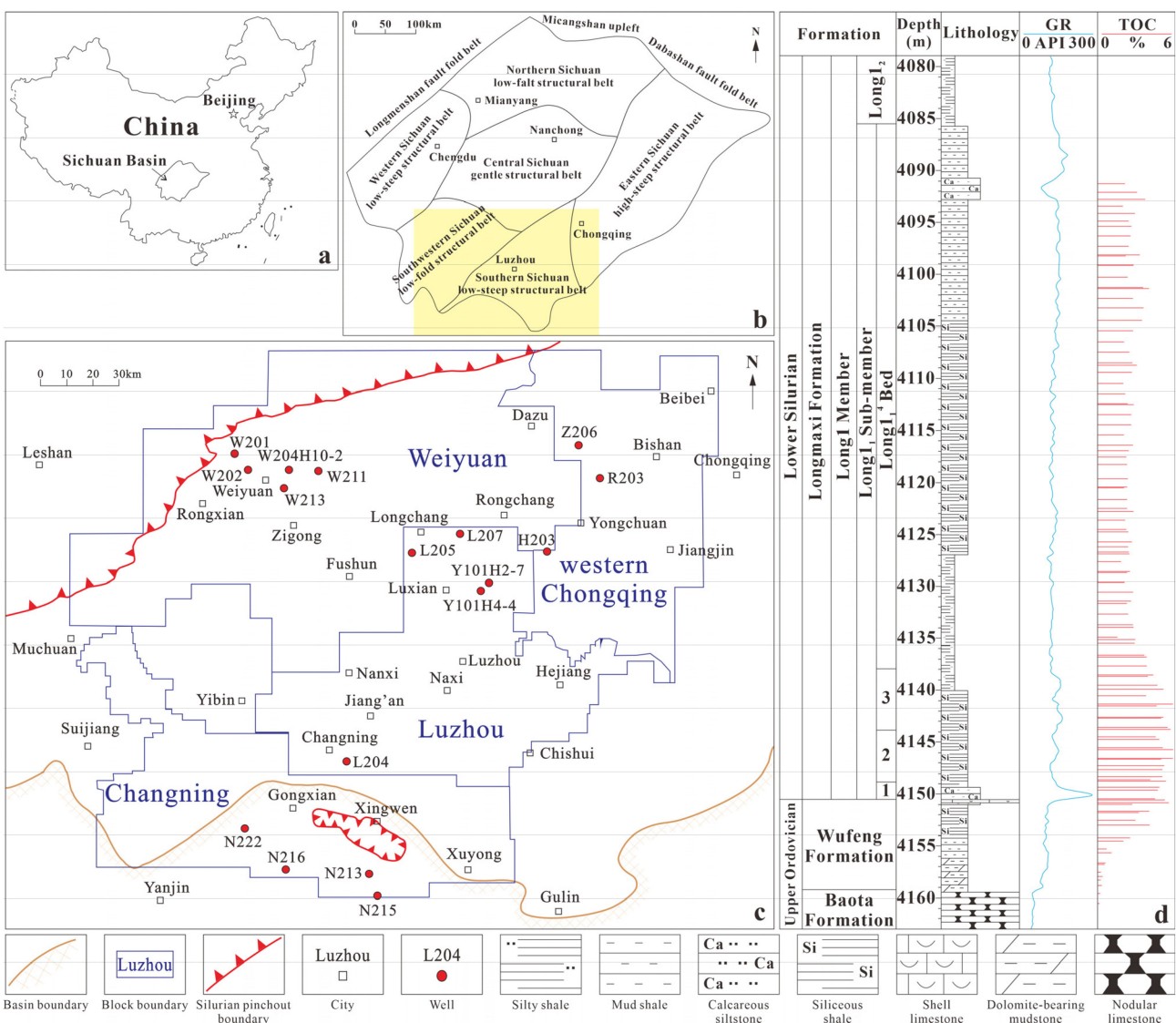

**Figure 1.** Location of the study area and wells (**a**–**c**), and the Wufeng Formation–$Long1_1$ sub-member stratigraphic column of Y101H4-4 well (**d**).

## 3. Samples and Methods

### 3.1. Samples

The samples consisted of cores taken from 17 wells in the Wufeng Formation to the $Long1_1$ sub-member, in different parts of the southern Sichuan Basin, with burial depths ranging from 1540 m to 4340 m. The locations of the sampled wells are shown in Figure 1. A total of 33 samples were taken, comprising 1–4 samples from each well according to the thickness of the Wufeng Formation to teh $Long1_1$ sub-member. The lithology of the samples is mainly black siliceous shale.

*3.2. Methods*

3.2.1. Organic Petrography and Optical Reflectance

The samples were cut perpendicular to bedding into approximately 2 cm × 2 cm blocks. In addition, for comparison, 10 samples from different wells were cut parallel to bedding into blocks of the same size. The samples were then cold set into an epoxy resin mixture and ground and polished after hardening. Organic petrology was carried out on selected samples using polished whole rock blocks. A Zeiss Axio Scope A1 digital microscopy with a J & M MSP 200 photometer was used to observe the optical properties of graptolites and solid bitumen and to test their random reflectance. This analysis was conducted with a ×50 oil-immersion objective using reflected light; the wavelength of the photometer was 546 nm and the room temperature was 26 °C. The reflectance measurement system was linearly calibrated against three known reflectance standards: Saphir ($R_o$ = 0.589%), Gadolinium–Gallium–Garnet ($R_o$ = 1.717%) and Cubic-zirconia ($R_o$ = 3.16%). Random reflectance measurement followed the industry standard SY/T 5124-2012. Samples were prepared and analyzed in the Geochemical Laboratory at Yangtze University.

3.2.2. Raman Spectroscopy of OM

A certain amount of shale sample from each core sample was weighed to isolate kerogen. The kerogen was isolated according to the national standard GB/T 19144-2010. A small amount of kerogen powder was placed on a slide, and the Raman spectroscopic experiments were then conducted on a Renishaw in Via Raman spectrometer after flattening and compacting the powder. A silicon wafer was used for wavenumber calibration before testing. An argon ion laser was excited with a 532 nm laser, which was focused on the maceral surface with a ×50 magnification objective, resulting in a spot size of 2 μm. The laser power was set to 2 mW and the grating was set to 1800 lines/mm. Five Raman spectra were collected randomly for each sample. Spectra were acquired over the range of 900–2000 cm$^{-1}$, while curve fitting and baseline correction were made by performing a Savitzky–Golay smoothing using a 21-point quadratic polynomial algorithm, a 3rd-order polynomial background correction, and normalization of the spectra to the same G intensity counts of 2000.

3.2.3. Basin Modelling

Based on the stratigraphic age, stratigraphic thickness, lithology, erosion amount, strata denudation process and thermal history, the burial and thermal maturity history of the Wufeng–Longmaxi Shale was reconstructed. The modelling of thermal maturity and burial history of the Wufeng–Longmaxi Shale were performed using PetroMod v11 software in 1D numerical mode. Using Raman spectroscopic parameters of organic matter as intermediate variables, the *EqVR*$_o$ calculated based on the *GR*$_{or}$ of the Wufeng–Longmaxi Formations from the southern Sichuan Basin were used to aid in calibrating maturation modelling. Maturation and hydrocarbon generation modelling were calculated by the EASY% *R*$_o$ method [52].

**4. Results**

*4.1. Optical Properties and Random Reflectance of Organic Matter*

The results of whole rock observation under the microscope show that graptolite and solid bitumen are the dominant forms of organic matter in the Wufeng–Longmaxi Formations. Accurate identification of graptolite and solid bitumen is the premise of accurate determination of thermal maturity.

4.1.1. Graptolites

The graptolites show two types of morphology under reflected light: granular and non-granular [20,53]. The non-granular graptolites mostly distribute along the bedding in strips, often appearing as segmented or discontinuous phenomena (Figure 2a,b). The non-granular graptolites have a smooth surface, high reflectance and obvious anisotropy. When

rotating the microscope stage 360°, the non-granular graptolites extinguish twice under polarized light. Some of them were still left with biological structures, such as walls and common canals, and the common canals often contain framboidal pyrite (Figure 2c). The fusellar layers on the parallel bedding sections are clearly seen when the polishing effect is good(Figure 2d); this phenomenon is more obvious with higher maturity. The granular graptolites are generally ellipsoidal and sometimes also distribute along the bedding. The surface of granular graptolite is rough, with an obvious sense of granularity (Figure 2e,f). The granular graptolite shows weak anisotropy.

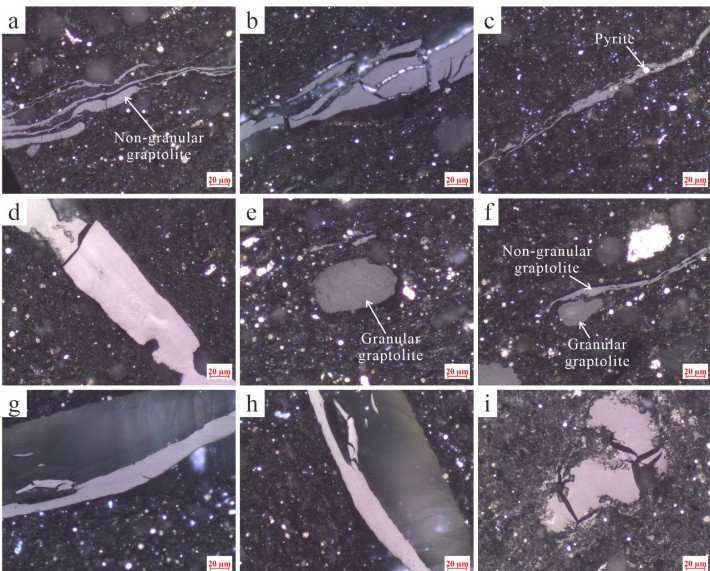

**Figure 2.** Microphotographs of graptolites in the Wufeng–Longmaxi Formations from southern Sichuan Basin. (**a**) Non-granular graptolite, distributing along the bedding in strips (L207, 3454.72 m, $GR_{or}$ = 3.327%); (**b**) Non-granular graptolite with intermittent form, distributing along the bedding (L207, 3454.72 m, $GR_{or}$ = 3.172%); (**c**) The common canal contains framboidal pyrite, non-granular graptolite (W202, 2568.20 m, $GR_{or}$ = 3.033%); (**d**) Fusellar layer of non-granular graptolite on parallel bedding section (L204, 3843.70 m, $GR_{or}$ = 5.310%); (**e**) Granular graptolite (L207, 3454.72 m, $GR_{or}$ = 1.794%); (**f**) Granular graptolite and non-granular graptolite (L207, 3454.72 m); (**g**) Non-granular graptolite (L204, 3825.79 m, $GR_{or}$ = 3.873%); (**h**) The same non-granular graptolite as (**g**), with 90° rotation of the microscope stage to show anisotropy ($GR_{or}$ = 3.356%). (**i**) Non-granular graptolite may be infected by bitumen (N222, 4327.14 m, $GR_{or}$ = 3.603%).

In general, granular graptolites and non-granular graptolites are found in carbonates and mudstones, respectively. They are derived from different sections of the rhabdosome. While the non-granular graptolites may be part of the wall with fusellar layers, the granular graptolites may be derived from the common canal [54,55]. In the Wufeng–Longmaxi Formations in the southern Sichuan Basin, the occurrence frequency of non-granular graptolites is significantly higher than that of granular graptolites. Therefore, the main test object of our study is the non-granular graptolites. Graptolite directional arrangement may exist during the sedimentary process. In the high–over-mature stage, the strong anisotropy of non-granular graptolites results in a significant difference in reflectance measured at different angles (Figure 2g,h). This may also cause the $GR_{or}$ obtained on the parallel bedding sections to be more scattered. Therefore, the random reflectance in this paper was obtained by testing different graptolites in the same direction with a fixed microscope stage. It must be noted that the graptolites may be infected by bitumen (Figure 2i), which results in a lower reflectance [53], so we ignored this type of graptolite.

### 4.1.2. Solid Bitumen

As secondary products, solid bitumen shows various models of accumulations in the Wufeng–Longmaxi Shale [23]. The source of solid bitumen is complex [22]. It can be a residue of kerogen cracking or a product of crude oil cracking, or it can also be formed by a combination of both [21,56]. Under the oil-immersion objective with reflected light, the reflection ability of solid bitumen is weak and there are no biological structure characteristics. The anisotropy of solid bitumen is weak when rotating the microscope stage 360° under single-polarized light. According to the microstructure and optical properties, solid bitumen in the Wufeng–Longmaxi Formations in the southern Sichuan Basin can be divided into two types: Type A and Type B (Figure 3). Type A is granular-angular solid bitumen with high reflectance, which fills and distributes in the pores (Figure 3a,b) or the edges of mineral particles (Figure 3c,d). Type A has smooth surface, obvious shape, sharp outline and mosaic structure (Figure 3c,e). Type B is matrix solid bitumen. Its surface is rough and shows amorphous shapes such as diffuse, filamentous, strip and floccule (Figure 3f–h). Type B is mainly disseminated in clay matrix, cracks or pores. Type B particles can form an aggregate of very fine microsomal particles (Figure 3i). Type B may be the secondary component produced by the continuous pyrolysis of heavy oil generated in the early stage, while Type A may have been formed before oil generation. Thus, Type A has experienced a more complete thermal evolution process, which can better reflect the degree of thermal evolution of OM. The accumulation pattern of solid bitumen is closely related to lithology [19,32]. Type B is abundant in clay-rich shale and is mostly dispersed in clay matrix. It shows the phenomenon of porosity and mineral interference, which may lead to a large error in the reflectance. Therefore, this paper selected Type A as the main test object of solid bitumen reflectance ($BR_o$).

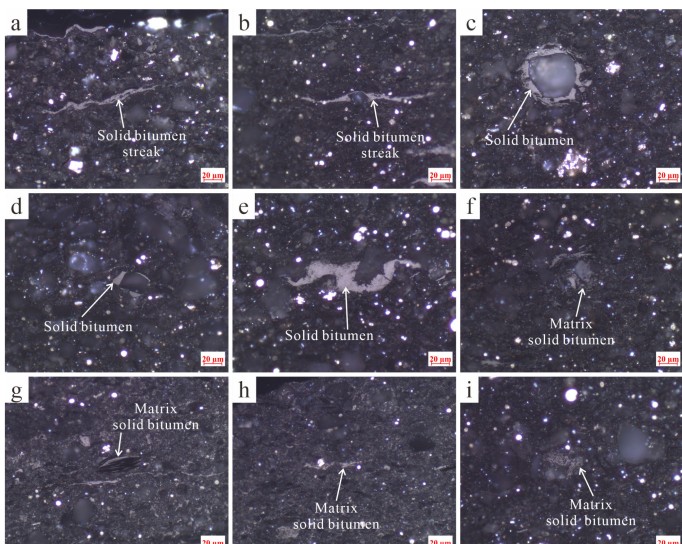

**Figure 3.** Microphotographs of solid bitumen in the Wufeng–Longmaxi Formations from southern Sichuan Basin. (**a**) Type A, filled in pores (L204-34, 3843.70 m, $BR_o$ = 2.687%); (**b**) Type A, filled in pores (Y101H4-4-44, 4142.63 m, $BR_o$ = 3.091%) (**c**) Type A, attached to the edge of mineral grain (L205-26, 4034.82 m, $BR_o$ = 3.031%); (**d**) Type A, attached to the edge of mineral grain (Y101H4-4-30, 4133.96 m, $BR_o$ = 2.213%); (**e**) Type A, showing mosaic structure (L204-17, 3825.79 m, $BR_o$ = 3.291%); (**f**) Type B, matrix solid bitumen cracked into angular particles with high reflectance (W213-1, 3745.34 m, $BR_o$ = 2.847%); (**g**) Type B, distributed in the matrix pores (N222-4, 4327.14 m); (**h**) Type B, distributed in the matrix pores (N222-4, 4327.14 m, $BR_o$ = 2.673%). (**i**) Aggregate formed by Type B particles (L204-17, 3825.79 m, $BR_o$ = 1.461%).

### 4.1.3. Random Reflectance of Graptolites and Solid Bitumen

According to previous research [57], compared with the graptolite maximum reflectance, the standard deviation (SD) of $GR_{or}$ is smaller, the value distribution is more

concentrated and the testing process is easier. Therefore, in this study, we measured the $GR_{or}$ of non-granular graptolites on the sections perpendicular to bedding and parallel to bedding. At the same time, we measured the $BR_o$ of the sections perpendicular to bedding.

Table 1 shows the average $GR_{or}$ of non-granular graptolites and solid bitumen in this study. On sections perpendicular to bedding, the $GR_{or}$ of the Wufeng–Longmaxi Shale in the southern Sichuan Basin ranges from 2.43% to 4.65%. The western Changning area has the highest $GR_{or}$, ranging from 3.77% to 4.65%, with an average of 4.13%. In Luzhou–western Chongqing, the $GR_{or}$ ranges from 2.97% to 3.64%, with an average of 3.34%. The $GR_{or}$ of eastern Changning ranges from 3.14% to 3.32%, with an average of 3.24%. The Weiyuan area has the lowest $GR_{or}$, ranging from 2.43% to 3.18%, with an average of 2.84%. On sections parallel to bedding, the $GR_{or}$ ranges from 2.88% to 5.58%, which is obviously higher than that on the vertical bedding sections. Three wells were selected to compare the $GR_{or}$ on sections parallel to bedding and perpendicular to bedding (Figure 4). The results show that the $GR_{or}$ on the sections parallel to bedding is much higher than that on the sections perpendicular to bedding, and the SD of $GR_{or}$ on the sections perpendicular to bedding is lower.

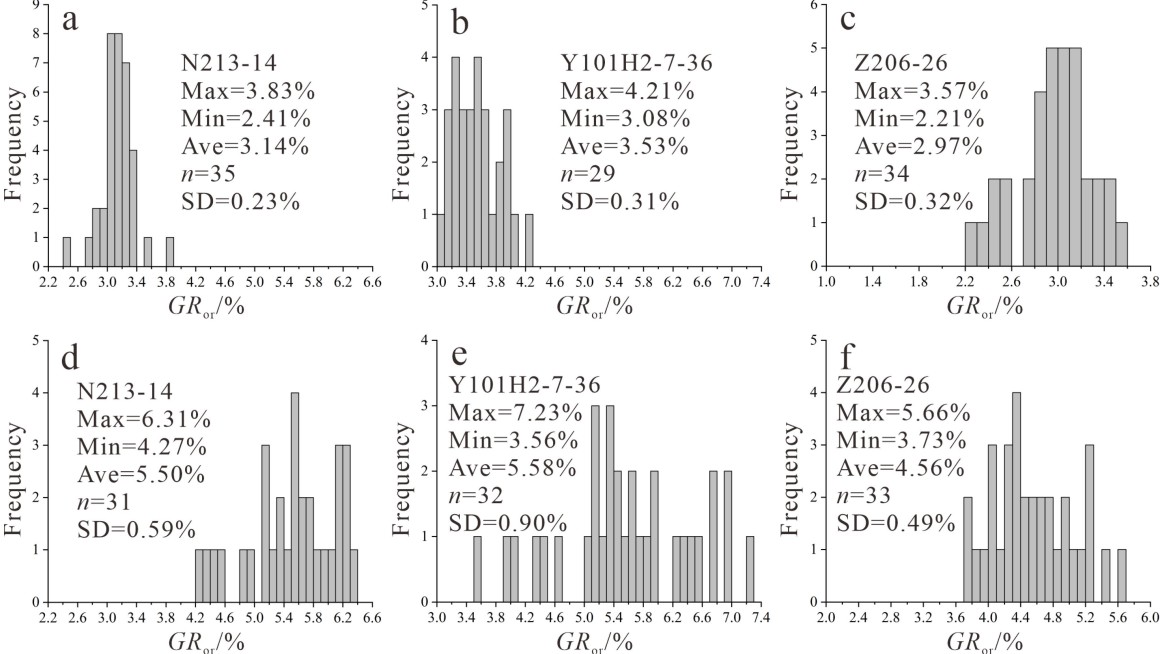

**Figure 4.** Comparison of the graptolite random reflectance of the Wufeng–Longmaxi Formations on sections perpendicular to bedding (**a–c**) and sections parallel to bedding (**d–f**) from southern Sichuan Basin.

The average $BR_o$ of Type A in the Wufeng–Longmaxi Formations in the southern Sichuan Basin ranges from 2.11% to 3.56%, which is lower than that of non-granular graptolites (Table 1, Figure 5). The western Changning area has the highest $BR_o$, ranging from 2.65% to 3.56%, with an average of 3.10%. In Luzhou-western Chongqing, $BR_o$ ranges from 2.31% to 3.15%, with an average of 2.73%. The $BR_o$ of eastern Changning ranges from 2.48% to 2.83%, with an average of 2.69%. The Weiyuan area has the lowest $BR_o$, ranging from 2.11% to 2.53%, with an average of 2.39%. The $BR_o$ of Type A showed a good relationship with the $GR_{or}$ (Figure 6; $R^2 = 0.69$). With the increase in maturity, the growth rate of $GR_{or}$ is higher than the $BR_o$ of Type A. Both the $GR_{or}$ and the $BR_o$ of Type A can be used to evaluate the maturity of Lower Paleozoic shale; however, compared with the $BR_o$ of Type A, the $GR_{or}$ is more concentrated in the sections perpendicular to bedding (Figure 5). Therefore, the $GR_{or}$ is more suitable as an indicator of thermal maturity.

**Table 1.** Random reflectance and equivalent vitrinite reflectance for graptolites and solid bitumen in the Wufeng–Longmaxi Formations from southern Sichuan Basin.

| Sample | Depth (m) | Formation | Perpendicular | | | $GR_{or}$ (Parallel, %) | $BR_o$ | | | $EqVR_o$-1 (%) | $EqVR_o$-2 (%) | $EqVR_o$-3 (%) | $EqVR_o$-4 (%) | $RmcR_o$ (%) | $EqVR_o$-5 (%) |
|---|---|---|---|---|---|---|---|---|---|---|---|---|---|---|---|
| | | | $GR_{or}$ (%) | SD | $n$ | | Type A (%) | SD | $n$ | | | | | | |
| Y101H4-4-30 | 4131.55 | Long$1_1^4$ | 3.52 | 0.41 | 18 | | 3.03 | 0.38 | 29 | 3.56 | 3.32 | 2.38 | 3.22 | 3.53 | 3.53 |
| Y101H4-4-44 | 4140.22 | Long$1_1^3$ | 3.58 | 0.29 | 21 | 5.49 | 2.91 | 0.28 | 15 | 3.63 | 3.34 | 2.29 | 3.06 | 3.56 | 3.55 |
| Y101H4-4-52 | 4144.14 | Long$1_1^2$ | 3.64 | 0.36 | 22 | | 2.88 | 0.29 | 12 | 3.69 | 3.35 | 2.28 | 3.03 | 3.53 | 3.57 |
| Y101H4-4-68 | 4150.6 | Wufeng | 3.53 | 0.23 | 34 | | 2.98 | 0.52 | 11 | 3.58 | 3.33 | 2.34 | 3.15 | 3.53 | 3.54 |
| Y101H2-7-29 | 4139.05 | Long$1_1^3$ | 3.45 | 0.33 | 33 | | 3.00 | 0.13 | 12 | 3.49 | 3.14 | 2.36 | 3.18 | 3.24 | 3.37 |
| Y101H2-7-36 | 4146.67 | Long$1_1^1$ | 3.53 | 0.31 | 29 | 5.58 | 3.15 | 0.37 | 15 | 3.58 | 3.33 | 2.46 | 3.38 | 3.53 | 3.54 |
| Z206-7 | 4244.57 | Long$1_1^4$ | 3.24 | 0.15 | 18 | | 2.43 | 0.30 | 20 | 3.29 | 2.95 | 1.97 | 2.45 | 3.02 | 3.18 |
| Z206-26 | 4266.76 | Long$1_1^1$ | 2.97 | 0.32 | 34 | 4.56 | 2.47 | 0.38 | 38 | 3.02 | 2.68 | 2.00 | 2.50 | 3.20 | 2.92 |
| Z206-38 | 4274.36 | Wufeng | | | | | | | | | | | | 3.22 | |
| L204-17 | 3824.3 | Long$1_1^3$ | 3.23 | 0.30 | 35 | | 2.71 | 0.52 | 31 | 3.28 | 2.93 | 2.16 | 2.80 | 3.31 | 3.17 |
| L204-34 | 3842.21 | Long$1_1^1$ | 3.50 | 0.25 | 16 | 5.49 | 2.70 | 0.26 | 15 | 3.55 | 3.32 | 2.15 | 2.79 | 3.55 | 3.53 |
| L205-20 | 4024.08 | Long$1_1^3$ | 3.25 | 0.14 | 15 | | 2.57 | 0.42 | 15 | 3.29 | 2.95 | 2.06 | 2.62 | 3.25 | 3.18 |
| L205-26 | 4034.44 | Wufeng | 3.37 | 0.10 | 17 | 4.57 | 2.81 | 0.28 | 14 | 3.42 | 3.07 | 2.23 | 2.94 | 3.22 | 3.30 |
| L207-30 | 3448.05 | Long$1_1^4$ | 3.11 | 0.06 | 3 | | 2.31 | 0.47 | 32 | 3.15 | 2.81 | 1.89 | 2.30 | 2.91 | 3.05 |
| L207-32 | 3450.92 | Long$1_1^3$ | 3.05 | 0.13 | 33 | 3.46 | 2.49 | 0.34 | 29 | 3.10 | 2.76 | 2.01 | 2.52 | 2.93 | 3.00 |
| L207-36 | 3457.41 | Wufeng | 3.07 | 0.15 | 11 | | 2.78 | 0.17 | 5 | 3.11 | 2.77 | 2.21 | 2.90 | 2.97 | 3.01 |
| N213-14 | 2575.28 | Long$1_1^2$ | 3.14 | 0.23 | 35 | 5.50 | 2.48 | 0.55 | 3 | 3.18 | 2.84 | 2.01 | 2.51 | 3.26 | 3.08 |
| N213-17 | 2579.22 | Wufeng | 3.32 | 0.27 | 29 | | 2.83 | 0.49 | 18 | 3.36 | 3.02 | 2.24 | 2.96 | 3.31 | 3.25 |
| N216-7 | 2311.78 | Long$1_1^3$ | 3.77 | 0.17 | 31 | | 2.97 | 0.33 | 21 | 3.81 | 3.38 | 2.34 | 3.14 | 3.57 | 3.61 |
| N216-10 | 2320.68 | Long$1_1^1$ | 3.80 | 0.13 | 24 | 4.97 | 2.65 | 0.43 | 11 | 3.84 | 3.39 | 2.12 | 2.72 | 3.56 | 3.62 |
| W204H10-2-15 | 3350.6 | Long$1_1^2$ | 2.84 | 0.25 | 33 | | 2.45 | 0.30 | 18 | 2.89 | 2.55 | 1.98 | 2.47 | 2.97 | 2.79 |
| W204H10-2-23 | 3355.3 | Long$1_1^1$ | 2.86 | 0.15 | 30 | | 2.53 | 0.29 | 31 | 2.91 | 2.57 | 2.04 | 2.58 | 2.84 | 2.82 |
| H203-4 | 3758.1 | Wufeng | 3.17 | 0.22 | 9 | | | | | 3.22 | 2.88 | | | 3.21 | 3.11 |
| W201-2 | 1541 | Long$1_1^1$ | 2.58 | 0.25 | 27 | 2.88 | 2.11 | 0.27 | 11 | 2.63 | 2.30 | 1.75 | 2.05 | 2.18 | 2.55 |
| W201-3 | 1548.8 | Wufeng | 2.43 | 0.25 | 4 | | | | | 2.48 | 2.15 | | | 2.41 | 2.41 |
| W202-1 | 2561.5 | Long$1_1$ | 3.18 | 0.22 | 35 | | 2.40 | 0.35 | 11 | 3.23 | 2.88 | 1.95 | 2.42 | 3.15 | 3.12 |
| W202-2 | 2568.2 | Long$1_1$ | 2.90 | 0.18 | 33 | | 2.53 | 0.41 | 14 | 2.95 | 2.62 | 2.04 | 2.57 | 2.70 | 2.86 |
| W213-1 | 3745.34 | Long$1_1$ | 3.02 | 0.22 | 35 | | 2.32 | 0.44 | 26 | 3.07 | 2.73 | 1.89 | 2.31 | 2.86 | 2.97 |
| W211-2 | 3559.5 | Long$1_1$ | 2.93 | 0.26 | 26 | | 2.39 | 0.45 | 12 | 2.98 | 2.64 | 1.94 | 2.40 | 3.09 | 2.88 |
| N222-1 | 4301.6 | Long$1_1$ | 4.65 | 0.59 | 24 | | 3.56 | 0.49 | 8 | 4.69 | 3.57 | 2.74 | 3.94 | 3.88 | 3.91 |
| N222-4 | 4327.14 | Long$1_1$ | 4.28 | 0.29 | 16 | 3.96 | 3.20 | 0.40 | 18 | 4.32 | 3.49 | 2.49 | 3.45 | 3.82 | 3.78 |
| N215-1 | 2502.95 | Long$1_1$ | 3.27 | 0.36 | 31 | | 2.75 | 0.36 | 19 | 3.32 | 2.97 | 2.19 | 2.85 | 3.38 | 3.20 |
| R203-1 | 4336.14 | Long$1_1$ | 3.55 | 0.44 | 23 | | 2.42 | 0.33 | 22 | 3.60 | 3.33 | 1.96 | 2.43 | 3.48 | 3.54 |

Note: $GR_{or}$: graptolite random reflectance; $BR_o$: solid bitumen random reflectance; SD: standard deviation; n: number of measuring points; $EqVR_o$: equivalent vitrinite reflectance; $RmcR_o$: calculated reflectance by Raman spectroscopic parameters; $EqVR_o$-1 = 0.99$GR_{or}$ + 0.08 (Luo et al., 2020 [27]); $EqVR_o$-2 = 0.97$GR_{or}$ − 0.2 (2.19% < $GR_{or}$ < 3.5%), $EqVR_o$-2 = 0.22$GR_{or}$ + 2.55 ($GR_{or}$ > 3.5%) (Wang et al., 2019 [28]); $EqVR_o$-3 = 0.679$BR_o$ + 0.3195 ($BR_o$ < 5.0%) (Feng and Chen, 1988 [58]); $EqVR_o$-4 = ($BR_o$ + 0.2443)/1.0495 ($R_o$ < 5.0%) (Schoenherr et al., 2007 [59]); $RmcR_o$ = 0.0537$RBS$ − 11.21 ($R_o$ < 3.5%), $RmcR_o$ = 1.1659($I_D/I_G$) + 2.7588 ($R_o$ > 3.5%) (Liu et al., 2013 [35]), the range of $R_o$ is divided by $EqVR_o$-1; $EqVR_o$-5 is the comprehensive evaluation result of this study.

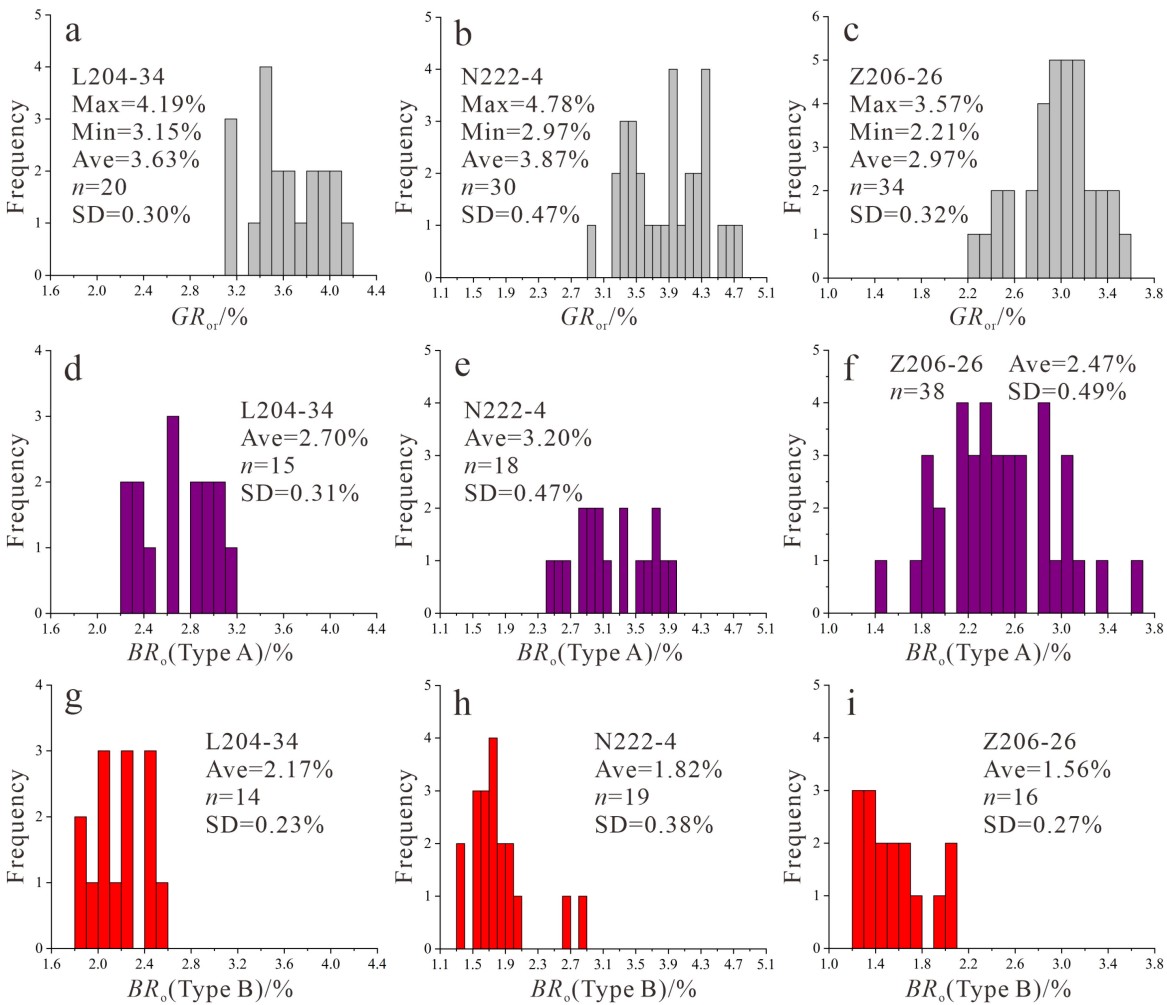

**Figure 5.** Comparison of the random reflectance of non-granular graptolites (**a–c**) and solid bitumen (Type A: **d–f**; Type B: **g–i**) in samples L204-34, N222-4 and Z206-26 on sections perpendicular to bedding. Max: maximum value; Min: minimum value; Ave: average value; n: number of measuring points; SD: standard deviation.

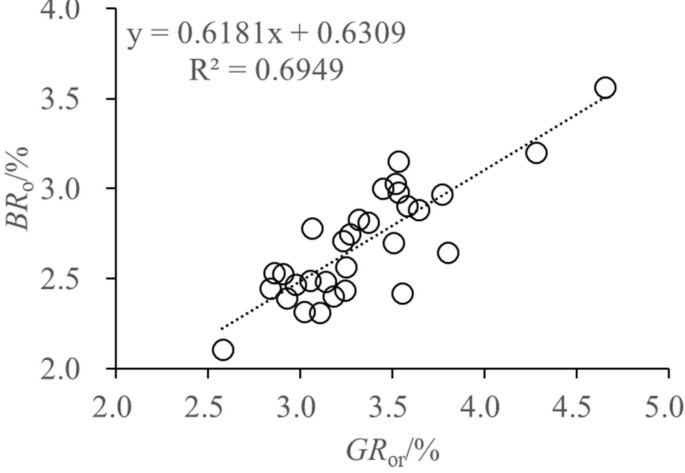

**Figure 6.** The relationship between the random reflectance of graptolites and solid bitumen (Type A) in the Wufeng–Longmaxi Formations from southern Sichuan Basin.

### 4.2. Raman Spectra of OM

The Raman spectrum of OM comprises a first-order region and a second-order region. For OM, the Raman spectrum is focused on the first-order region, which comprises two main bands. The G band (graphite peak) is located at about 1600 cm$^{-1}$ and belongs to the C=C atomic bond on the aromatic structure plane. The position at about 1350 cm$^{-1}$ reflects the disorder of organic matter and the defect of structural units and is known as the D band (defect peak). In the process of thermal evolution, the chemical structure of OM changes regularly, and the corresponding Raman spectroscopic parameters also change regularly. The position of the D peak ($W_D$) and the G peak ($W_G$), the band separation between the D band and the G band (*RBS*), the area ratio of the D band and the G band ($A_D/A_G$), the intensity ratio between D and G bands ($I_D/I_G$) and the full width at half maximum of the G band (G-*FWHM*) and D band (D-*FWHM*) are increasingly used by scholars to quantitatively calculate the maturity of OM [35,36,60].

The Raman spectroscopic parameters of OM measured in this paper are shown in Table 2. In the first-order region, the D peak is in the range of 1327–1347 cm$^{-1}$ and the G peak lies in the range of 1595–1603 cm$^{-1}$; the G peak is narrower and sharper than the D peak. As shown in Figure 7, the G band and D band become narrower and sharper as maturity increases. Due to this change, D-*FWHM* decreases from 206.09 cm$^{-1}$ to 100.83 cm$^{-1}$, and G-*FWHM* decreases from 77.92 cm$^{-1}$ to 56.76 cm$^{-1}$, displaying a negative correlation with $GR_{or}$.

**Table 2.** Raman spectroscopic parameters of kerogen in the Wufeng–Longmaxi Formations from southern Sichuan Basin.

| Sample | $GR_{or}$ (%) | $W_G$ (cm$^{-1}$) | $W_D$ (cm$^{-1}$) | *RBS* (cm$^{-1}$) | $I_D/I_G$ | D-*FWHM* (cm$^{-1}$) | G-*FWHM* (cm$^{-1}$) |
|---|---|---|---|---|---|---|---|
| Y101H4-4-30 | 3.52 | 1599.54 | 1328.13 | 271.41 | 0.66 | 147.70 | 51.55 |
| Y101H4-4-44 | 3.58 | 1598.57 | 1328.13 | 270.44 | 0.68 | 156.71 | 58.05 |
| Y101H4-4-52 | 3.64 | 1599.21 | 1328.13 | 271.08 | 0.66 | 147.39 | 50.57 |
| Y101H4-4-68 | 3.53 | 1599.54 | 1328.47 | 271.07 | 0.66 | 149.05 | 51.55 |
| Y101H2-7-29 | 3.45 | 1599.21 | 1330.15 | 269.06 | 0.68 | 150.35 | 53.50 |
| Y101H2-7-36 | 3.53 | 1600.19 | 1330.49 | 269.70 | 0.66 | 149.77 | 50.90 |
| Z206-7 | 3.24 | 1598.24 | 1333.18 | 265.06 | 0.64 | 185.68 | 58.70 |
| Z206-26 | 2.97 | 1599.21 | 1330.82 | 268.39 | 0.63 | 167.60 | 53.18 |
| Z206-38 | | 1601.48 | 1332.84 | 268.64 | 0.62 | 155.81 | 49.59 |
| L204-17 | 3.23 | 1598.89 | 1328.47 | 270.42 | 0.69 | 149.35 | 53.82 |
| L204-34 | 3.50 | 1599.54 | 1330.15 | 269.39 | 0.67 | 148.37 | 52.53 |
| L205-20 | 3.25 | 1602.45 | 1333.18 | 269.28 | 0.60 | 160.88 | 48.61 |
| L205-26 | 3.37 | 1597.92 | 1329.14 | 268.78 | 0.65 | 170.95 | 55.13 |
| L207-30 | 3.11 | 1598.57 | 1335.53 | 263.03 | 0.60 | 193.35 | 59.35 |
| L207-32 | 3.05 | 1597.59 | 1334.19 | 263.41 | 0.61 | 197.42 | 59.04 |
| L207-36 | 3.07 | 1597.92 | 1333.85 | 264.07 | 0.61 | 193.00 | 59.68 |
| N213-14 | 3.14 | 1599.86 | 1330.49 | 269.38 | 0.65 | 141.31 | 49.27 |
| N213-17 | 3.32 | 1598.89 | 1328.47 | 270.42 | 0.71 | 146.37 | 52.53 |
| N216-7 | 3.77 | 1599.54 | 1327.46 | 272.08 | 0.70 | 139.27 | 51.55 |
| N216-10 | 3.80 | 1601.81 | 1331.50 | 270.31 | 0.69 | 150.74 | 50.89 |
| W204H10-2-15 | 2.84 | 1597.92 | 1333.85 | 264.07 | 0.62 | 199.43 | 60.98 |
| W204H10-2-23 | 2.86 | 1597.59 | 1335.87 | 261.73 | 0.63 | 200.11 | 64.89 |
| H203-4 | 3.17 | 1597.92 | 1329.48 | 268.44 | 0.64 | 176.66 | 57.41 |
| W201-2 | 2.58 | 1595.97 | 1346.62 | 249.36 | 0.61 | | 75.31 |
| W201-3 | 2.43 | 1596.30 | 1342.59 | 253.71 | 0.61 | | 70.74 |
| W202-1 | 3.18 | 1600.51 | 1333.18 | 267.33 | 0.63 | 206.09 | 60.82 |
| W202-2 | 2.90 | 1598.89 | 1339.90 | 258.99 | 0.59 | 203.46 | 62.93 |
| W213-1 | 3.02 | 1598.24 | 1336.21 | 262.04 | 0.63 | 196.70 | 60.98 |

**Table 2.** *Cont.*

| Sample | $GR_{or}$ (%) | $W_G$ (cm$^{-1}$) | $W_D$ (cm$^{-1}$) | $RBS$ (cm$^{-1}$) | $I_D/I_G$ | D-FWHM (cm$^{-1}$) | G-FWHM (cm$^{-1}$) |
|---|---|---|---|---|---|---|---|
| W211-2 | 2.93 | 1599.86 | 1333.52 | 266.35 | 0.60 | 184.37 | 55.45 |
| N222-1 | 4.65 | 1598.89 | 1337.89 | 261.01 | 0.96 | 100.83 | 55.77 |
| N222-4 | 4.28 | 1598.57 | 1335.87 | 262.70 | 0.91 | 114.98 | 57.08 |
| N215-1 | 3.27 | 1601.48 | 1329.81 | 271.67 | 0.65 | 143.34 | 48.62 |
| R203-1 | 3.55 | 1601.81 | 1331.16 | 270.65 | 0.62 | 159.81 | 50.57 |

Note: $GR_{or}$: graptolite random reflectance; $W_G$: the position of G peak; $W_D$: the position of D peak; $RBS$: the band separation between D and G bands; $I_D/I_G$: the intensity ratio between D and G bands; D-FWHM: the full width at half maximum of D band; G-FWHM: the full width at half maximum of G band.

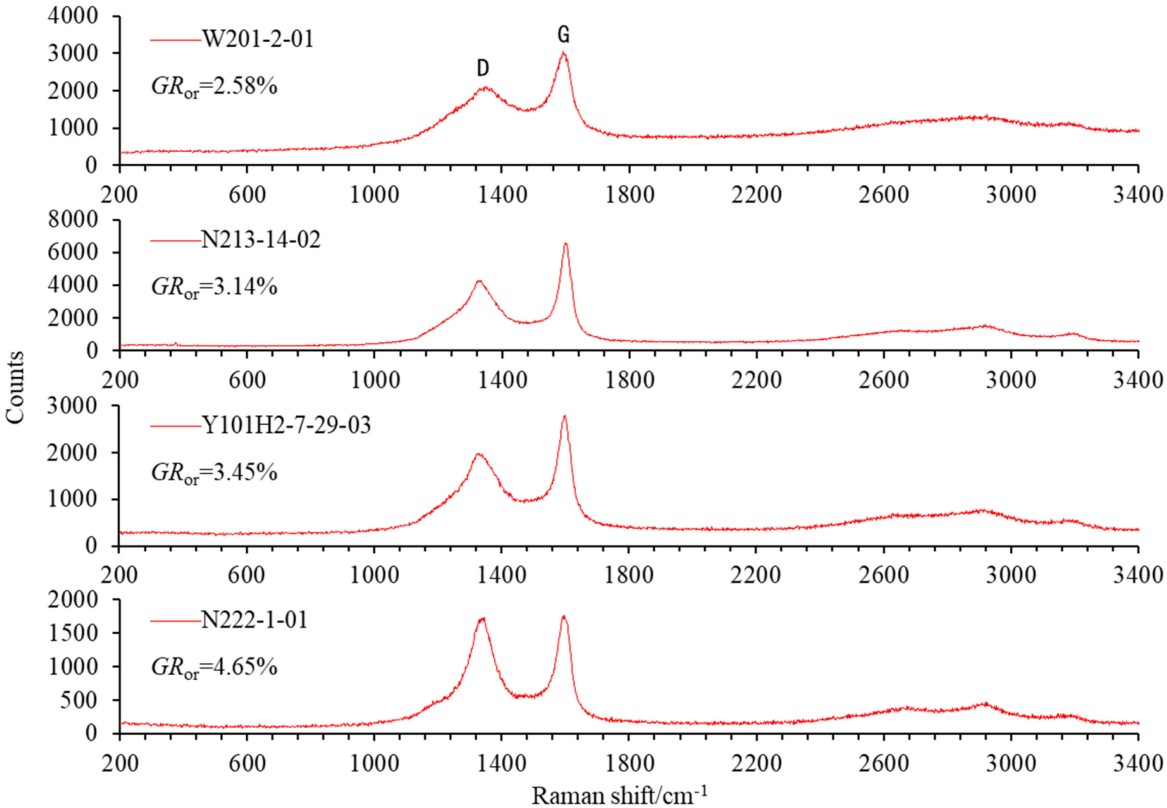

**Figure 7.** Raman spectra for solid organics with different maturities.

In the process of thermal evolution, the Raman spectroscopic parameters of OM from all 33 samples also show regular changes with the increase in the $GR_{or}$ (Figure 8). The D peak first shifted to lower wave numbers as the maturity increased, but then shifted to higher wave numbers, until the $GR_{or}$ increased beyond about 3.5% (Figure 8a). In contrast, the G peak displayed a small trend of higher wave numbers shifting at the beginning, and then stabilizing at around 1600 cm$^{-1}$ (Figure 8b). Under the control of $W_D$ and $W_G$, $RBS$ gradually increased. However, RBS no longer increased but decreased when $GR_{or}$ was in the range of 3.5%–5.0% (Figure 8c). With the increase in the $GR_{or}$, $I_D/I_G$ increased slightly, but $I_D/I_G$ showed an obvious increasing trend until the $GR_{or}$ increased beyond about 3.5% (Figure 8d). The D-FWHM trend decreased with increasing maturity (Figure 8e). The G-FWHM also decreased with the increase in the $GR_{or}$ at first, but when the $GR_{or}$ increased beyond about 3.5%, G-FWHM began to increase slightly (Figure 8f).

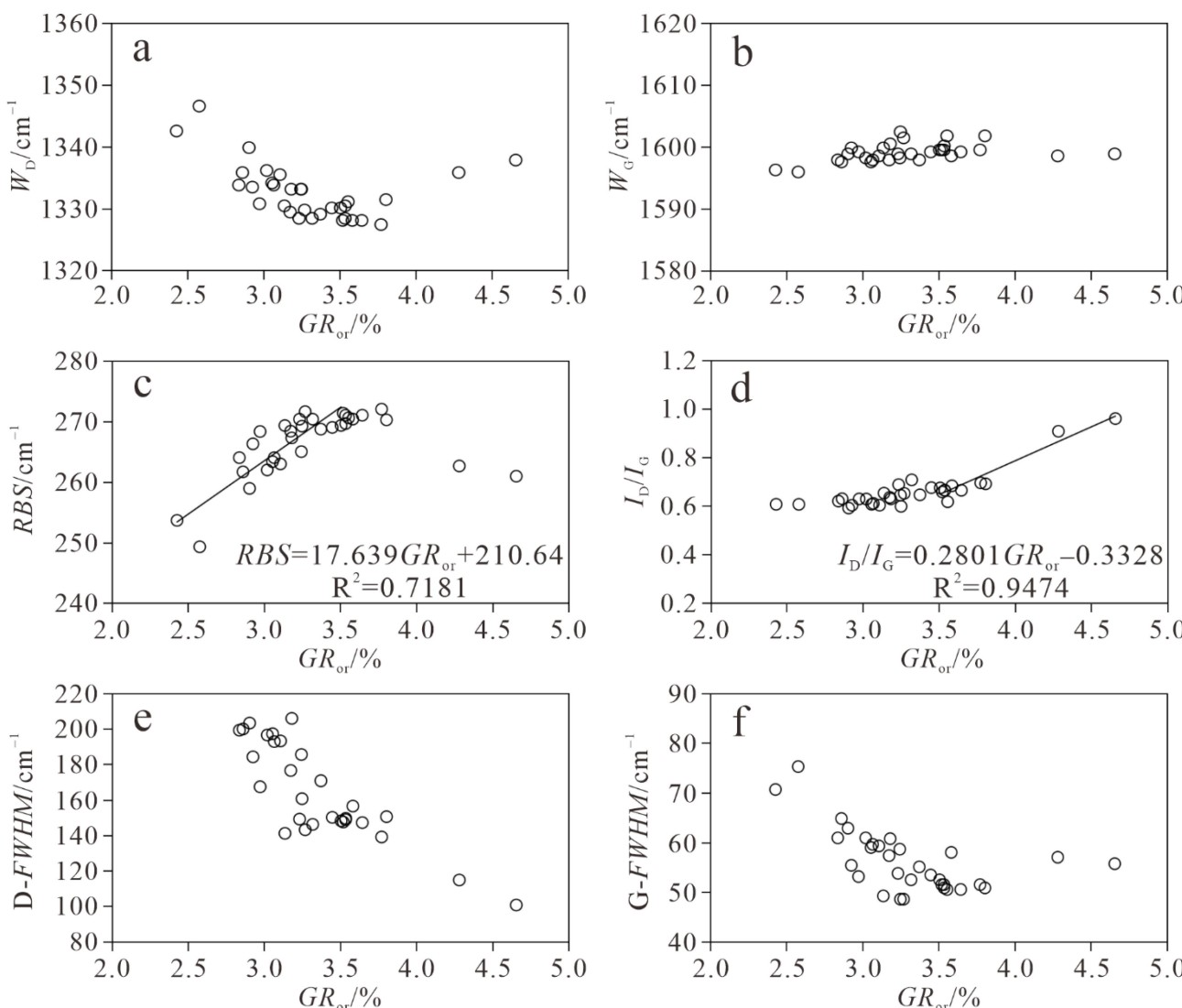

**Figure 8.** Plots of graptolite random reflectance versus Raman spectroscopic parameters of kerogen in the Wufeng–Longmaxi Formations from southern Sichuan Basin. (**a**) relationship between the $W_D$ and $GR_{or}$; (**b**) relationship between the $W_G$ and $GR_{or}$; (**c**) relationship between the $RBS$ and $GR_{or}$; (**d**) relationship between the $I_D/I_G$ and $GR_{or}$; (**e**) relationship between the D-*FWHM* and $GR_{or}$; (**f**) relationship between the G-*FWHM* and $GR_{or}$. Circle represents the Raman spectroscopic parameter of kerogen corresponding to different $GR_{or}$; line represents the linear relationship between $GR_{or}$ and Raman spectroscopic parameters of kerogen.

### 4.3. Thermal Evolution of the Wufeng–Longmaxi Formations

4.3.1. Erosion Amount and Heat Flow

The Sichuan Basin is a typical multi-cyclical superimposed basin, which experienced multiple periods of uplifting, erosion, burial and deposition. The main tectonic movements in the southern Sichuan Basin include the Guangxi Movement during the Caledonian period, the Dongwu Movement during the Hercynian period, the Indosinian Movement, the Yanshan Movement and the Himalayan movement [61]. Based on the strata division data obtained from actual drilling and the previous research results of erosion events [61–68], this paper analyzes the differences in erosion thickness in different tectonic movement periods in the southern Sichuan Basin. The Guangxi Movement during the Caledonian period resulted in the absence of upper Silurian in the Sichuan Basin, and the erosion thickness decreased from the Leshan–Longnüsi inherited paleo-uplift to surrounding

areas [62]. In the Caledonian period, the erosion thickness of the Weiyuan area reached as much as 800 m, and in other areas it was about 100–200 m. The Dongwu Movement lasted for a short time, and the uplift and erosion were controlled by the dome formed by the magma upwelling before the eruption of Emeishan basalt [63]. The erosion thickness in the southwest Sichuan Basin near the Emeishan mantle plume was greater than in other areas. According to the strata division data, the fourth member of the Lower Permian Maokou Formation in southern Sichuan was denuded, and the thickness of denudation was about 100m, which was consistent with the conclusion of Jiang et al. [64]. The Indosinian movement caused the formation of the Luzhou–Kaijiang paleo-uplift in the southeastern Sichuan basin, and the Middle Triassic Leikoupo Formation suffered erosion in a large area. In Luzhou–western Chongqing, the erosion thickness was about 200–300 m, and the Lower Triassic Jialingjiang Formation and the Upper Triassic Xujiahe Formation had unconformity contact. The Leikoupo Formation in the Weiyuan and Changning areas was not completely denuded, and the thickness of denudation was about 100–150 m. Since the Late Cretaceous, the erosion thickness was more than 3500 m in the Weiyuan area, less than 3500 m in the Changning area and about 1000~2000 m in Luzhou–western Chongqing due to the Himalayan movement.

In view of the characteristics of multi-stage and complex thermal histories of super-imposed basins, the thermal evolution history of the Sichuan Basin since the Late Sinian was reconstructed by various thermal indicators (e.g., vitrinite reflectance, solid bitumen reflectance and apatite fission track data) and geodynamic methods [67–72]. The results of previous studies show that the Sichuan Basin was a stable craton basin before the Permian, and the heat flow was low and stable at 52 mW/m$^2$–59 mW/m$^2$. The Emeishan mantle plume movement occurred on the western margin of the Yangtze plate in the Middle and Late Permian, and its influence area can be divided into three parts: inner zone, middle zone and outer zone [63,72]. The heat effect is weakened from inside to outside. Controlled by the Emeishan mantle plume and basin-extensional thinning, a peak of heat flow occurred in the Sichuan Basin during the Middle and Late Permian. The maximum heat flow decreased gradually from southwest to northeast, and the peak value of heat flow in the southern Sichuan Basin was 70 mW/m$^2$–80 mW/m$^2$. The west of the Changning area is located in the middle zone of the Emeishan mantle plume, while the Luzhou–western Chongqing and Weiyuan areas are located in the outer zone of the Emeishan mantle plume. Therefore, in the west of Changning, the thermal effect of Emeishan basalt is stronger than that in other areas, which may lead to abnormal thermal maturity. After the Triassic, the heat flow decreased gradually, and the current heat flow in the southern Sichuan Basin was 60 mW/m$^2$–65 mW/m$^2$. The heat flow history of the Sichuan Basin since the Late Silurian is shown in Figure 9.

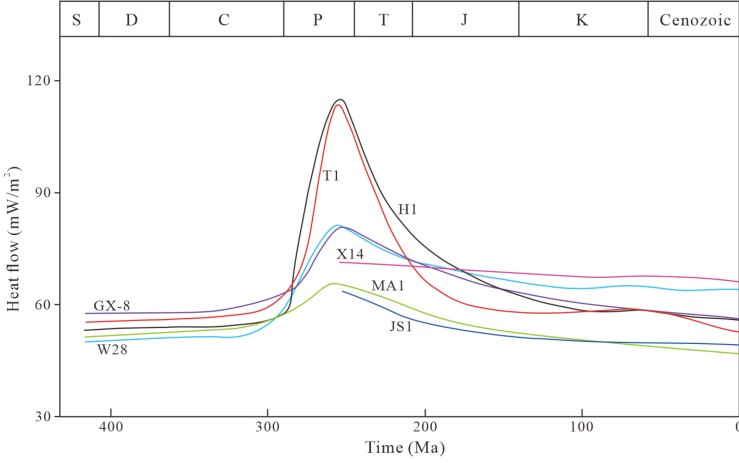

**Figure 9.** The heat flow evolution history since the Late Silurian of representative boreholes in the Sichuan Basin (according to Jiang et al., 2018 [72]).

#### 4.3.2. Burial and Thermal Evolution History

Based on the determination of erosion amounts and heat flows in different periods in the southern Sichuan Basin, the burial history and thermal evolution of the Wufeng–Longmaxi Formations in key wells were reconstructed by PetroMod V11 software. The results show that the Wufeng–Longmaxi Formations in the southern Sichuan Basin mainly experienced three burial stages, which occurred in the Silurian–Devonian, Middle Permian–Early Triassic and Late Triassic–Late Cretaceous. There are similarities and differences in the maturity evolution of different areas (Figure 10).

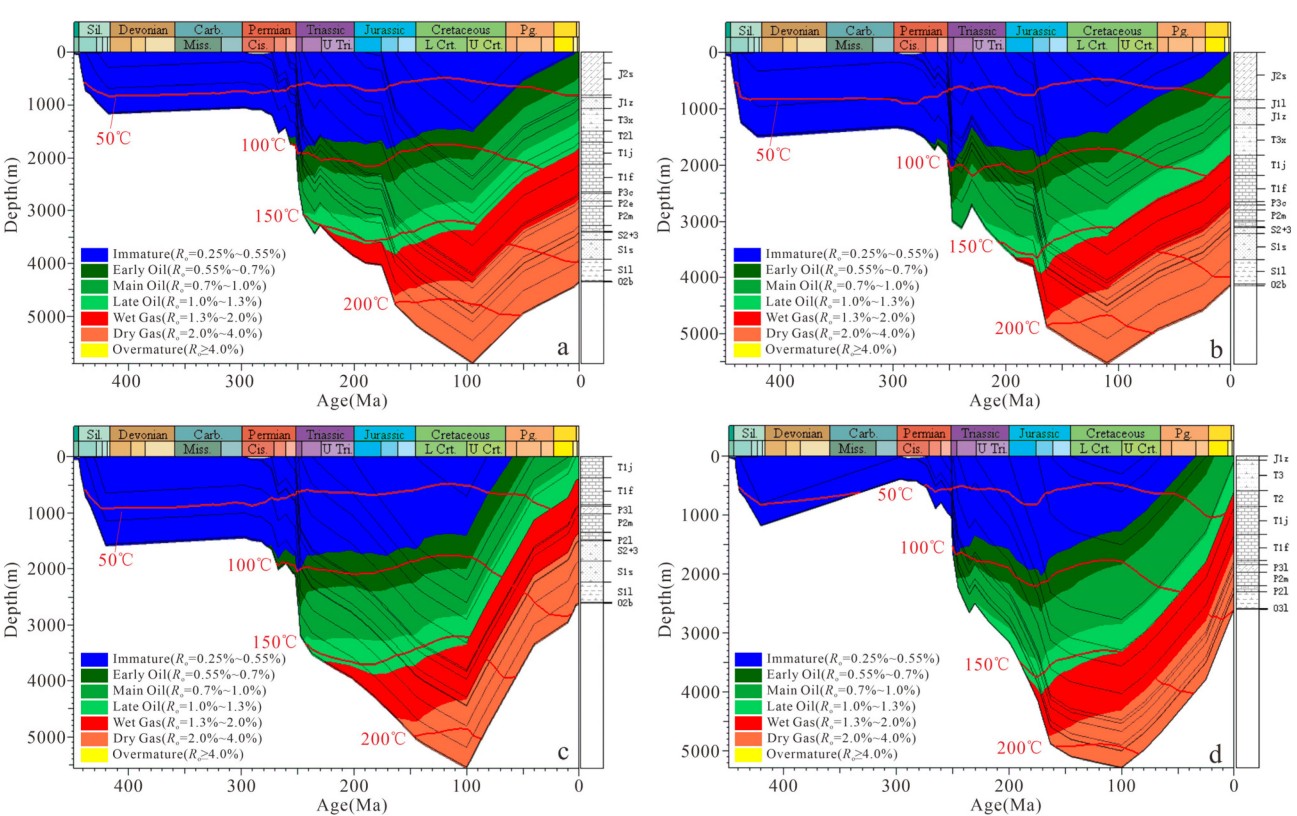

**Figure 10.** The burial, thermal history and maturation of the Wufeng–Longmaxi Formations in well N222 (**a**), Y101H4-4 (**b**), N213 (**c**) and W202 (**d**).

The burial depth of the Wufeng–Longmaxi Shale in the southern Sichuan Basin was less than 1500 m in the Late Silurian. It was immature at the end of the first burial stage, with $R_o$ < 0.5%. Except the Weiyuan area, other areas were only slightly uplifted during the Caledonian period, and the temperature changed little. From the Middle Permian–Early Triassic, controlled by the activity of extension, the Sichuan Basin rapidly settled and accepted deposition, and the Wufeng–Longmaxi Formations entered the second burial stage. The Dongwu Movement in the Middle and Late Permian led to a small-scale uplift, but it only lasted for a short time and had little effect on the thermal evolution of OM. In the early Middle Permian, the $R_o$ of the Wufeng–Longmaxi Formations in the Changning area and Luzhou–western Chongqing reached 0.55% and entered the hydrocarbon generation threshold. However, the time of hydrocarbon generation in the Weiyuan area was later than that in other areas and the $R_o$ of W202 did not reach 0.55% until the Early Triassic. This may have been caused by a larger erosion thickness and shallower burial depth in the Caledonian period. Under the influence of the Emeishan mantle plume thermal effect and basin stretching and thinning, the heat flow of the Sichuan Basin increased rapidly in the Middle and Late Permian. Influenced by the increase in burial depth and heat flow, the Wufeng–Longmaxi Shale matured rapidly. The maturation rate of the Changning area was significantly higher than that of other areas. At the end of the Middle Triassic, the $R_o$ of



Weiyuan and Luzhou–western Chongqing reached 1.0% and the formation temperature was about 130 °C. At this time, the $R_o$ of N222 and N213 wells, which were closer to the Emeishan mantle plume, had reached 1.3%, and the formation temperature was close to 150 °C. After the Late Triassic, the Wufeng–Longmaxi Formations entered the third burial stage and experienced a second rapid maturation due to the continuous increase in burial depth. The burial rate in the east of Changning was obviously lower than that in other areas, so the maturity increased slowly. In the Early Jurassic, the $R_o$ of W202 and Y101H4-4 wells reached 1.3%. The Wufeng–Longmaxi Formations in the southern Sichuan Basin were generally in a high maturity stage. In the Middle Jurassic, the $R_o$ reached 2.0%. At the end of the Early Cretaceous, the Wufeng–Longmaxi Formations in the southern Sichuan Basin reached the maximum burial depth and maturity. The maximum burial depths of the N222, Y101H4-4, N213 and W202 wells were 5885 m, 5510 m, 5540 m and 5275 m, respectively, and the corresponding $R_o$ was 3.75%, 3.34%, 3.18% and 3.03%. After the Late Cretaceous, the Sichuan Basin continued to be uplifted on a large scale, the formation temperature decreased and the maturity evolution of the Wufeng–Longmaxi Shale stopped.

4.3.3. Calibration of Thermal Maturity and Maximum Paleo-Temperature

The $R_o$ simulation values of single wells in different areas were calculated by using the 'Sweeney and Burnham (1990) EASY%$R_o$' organic-matter thermal-evolution kinetic model. Comparing the simulated values of $R_o$ to the actual $EqVR_o$ (Figure 11), the results show that they are in good agreement with each other, which proves that the simulation results of thermal evolution are credible.

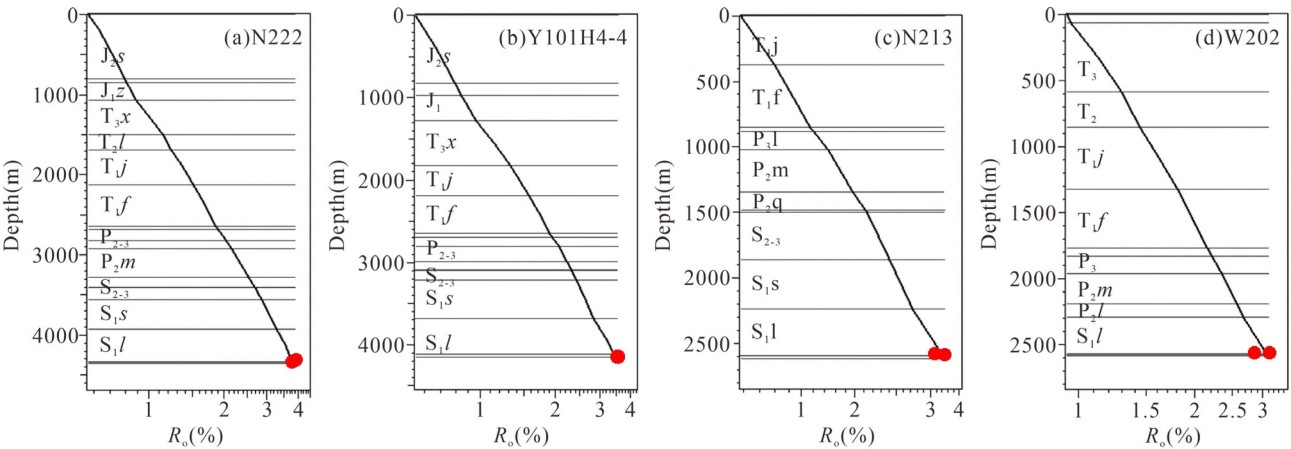

**Figure 11.** $R_o$ simulation value (curves) versus calculated $EqVR_o$ (red dots) for well N222 (**a**), Y101H4-4 (**b**), N213 (**c**) and W202 (**d**).

Barker and Pawlewicz [73] proposed that the vitrinite geological thermometer could be used to estimate the maximum paleo-temperature ($T_{\text{peak burial}}$) of the burial history (Equation (1)). In this paper, $EqVR_o$ was used to estimate the $T_{\text{peak burial}}$ of the Wufeng–Longmaxi Formations in the southern Sichuan Basin (Table 3).

$$T_{\text{peak burial}} \, (^\circ C) = (\ln R_o + 1.68)/0.0124 \qquad (1)$$

After conversion, the $T_{\text{peak burial}}$ of the two samples from the W202 well in the Weiyuan area were 227.27 °C and 220.17 °C, with an average of 223.72 °C. The $T_{\text{peak burial}}$ of the two samples from the N213 well in the east of the Changning area were 226.16 °C and 230.57 °C, with an average of 228.36 °C. The $T_{\text{peak burial}}$ in Luzhou–western Chongqing ranges from 222.00 °C to 238.16 °C, with an average of 230.98 °C. The $T_{\text{peak burial}}$ in western Changning ranges from 239.10 °C to 245.36 °C, with an average of 241.65 °C. From a horizontal point of view, the maximum paleo-temperature of the Wufeng–Longmaxi Formations in the Weiyuan, eastern Changning, Luzhou–western Chongqing and western Changning

areas increases successively. The maximum paleo-temperature of the Changning area increases from the N213 well to the N222 well near the Emeishan mantle plume. The average $T_{\text{peak burial}}$ of N213, N216 and N222 wells are 228.36 °C, 239.22 °C and 244.08 °C, respectively. The maximum paleo-temperature estimated by $EqVR_o$ is basically consistent with the maximum paleo-temperature simulated by the PetroMod V11 software, which indicates that the simulation results of thermal evolution are reliable.

**Table 3.** Maximum paleo-temperatures estimation.

| Sample | Depth (m) | $EqVR_o$ (%) | $T_1$ (°C) | Mean Value of $T_1$ (°C) | $T_2$ (°C) |
|---|---|---|---|---|---|
| Y101H4-4-30 | 4131.55 | 3.53 | 237.22 | | |
| Y101H4-4-44 | 4140.22 | 3.55 | 237.70 | | |
| Y101H4-4-52 | 4144.14 | 3.57 | 238.16 | 237.60 | 230.37 |
| Y101H4-4-68 | 4150.60 | 3.54 | 237.34 | | |
| L207-30 | 3448.05 | 3.05 | 225.43 | | |
| L207-32 | 3450.92 | 3.00 | 224.12 | 224.65 | 215.48 |
| L207-36 | 3457.41 | 3.01 | 224.40 | | |
| Z206-7 | 4244.57 | 3.18 | 228.80 | 225.40 | 224.20 |
| Z206-26 | 4266.76 | 2.92 | 222.00 | | |
| N213-14 | 2575.28 | 3.08 | 226.16 | 228.36 | 223.96 |
| N213-17 | 2579.22 | 3.25 | 230.57 | | |
| N216-7 | 2311.78 | 3.61 | 239.10 | 239.22 | 239.88 |
| N216-10 | 2320.68 | 3.62 | 239.34 | | |
| N222-1 | 4301.60 | 3.91 | 245.36 | 244.08 | 244.70 |
| N222-4 | 4327.14 | 3.78 | 242.79 | | |
| W202-1 | 2561.50 | 3.12 | 227.27 | 223.72 | 214.20 |
| W202-2 | 2568.20 | 2.86 | 220.17 | | |

Note: $T_1$ (°C) = (ln$R_o$ + 1.68)/0.0124 (Barker and Pawlewicz, 1994 [73]); $T_2$ (°C): the simulated maximum paleo-temperature.

## 5. Discussion

### 5.1. The Relationships between Raman Spectral Parameters and $GR_{or}$

Much research was conducted to establish the $EqVR_o$ conversion equation using different thermal evolution parameters [20,74–76]; however, these may not be applicable to the high–over-mature Wufeng-Longmaxi Formations in the Sichuan Basin. Luo et al. [27] combined published data with their study results of both natural and heat-treated graptolites and vitrinite, and presented a conversion equation between $GR_{or}$ and $EqVR_o$:

$$EqVR_o = 0.99GR_{or} + 0.08 \tag{2}$$

Liu et al. [35] proposed the two-stage linear regressions between vitrinite reflectance and Raman spectral parameters ($RBS$ and $I_D/I_G$) by using a 532 nm laser wavelength:

$$RmcR_o = 0.0537RBS - 11.21 \ (R_o < 3.5\%) \tag{3}$$

$$RmcR_o = 1.1659(I_D/I_G) + 2.7588 \ (R_o > 3.5\%) \tag{4}$$

Theoretically, the $EqVR_o$ converted from the $GR_{or}$ or Raman spectral parameters should be consistent or approximate. After substituting the $GR_{or}$ obtained in our study into Equation (2), a group of $EqVR_o$ was obtained, and the maturity was divided preliminarily. The corresponding Raman spectrum parameters were then substituted into Equations (3) and (4), and the second set of $EqVR_o$ ($RmcR_o$) was obtained. After comparing the two groups of conversion results, it was found that when the $GR_{or} < 3.5\%$, the $EqVR_o$ obtained by the two methods is close, and the difference is almost always within 0.3%. The result indicates that when $GR_{or} < 3.5\%$, the graptolites and vitrinite have a similar thermal evolution trend. However, when $GR_{or} > 3.5\%$, equivalent to $EqVR_o > 3.5\%$, vitrinite begins to graphitize, while graptolites maintain the original thermal evolution trend. At this stage,

the increasing rate of reflectance of vitrinite is higher than that of graptolite [28]. With the increase in the $GR_{or}$, the difference in conversion results between the two methods increases significantly (Figure 12), which is consistent with the results of previous studies [28,30,31].

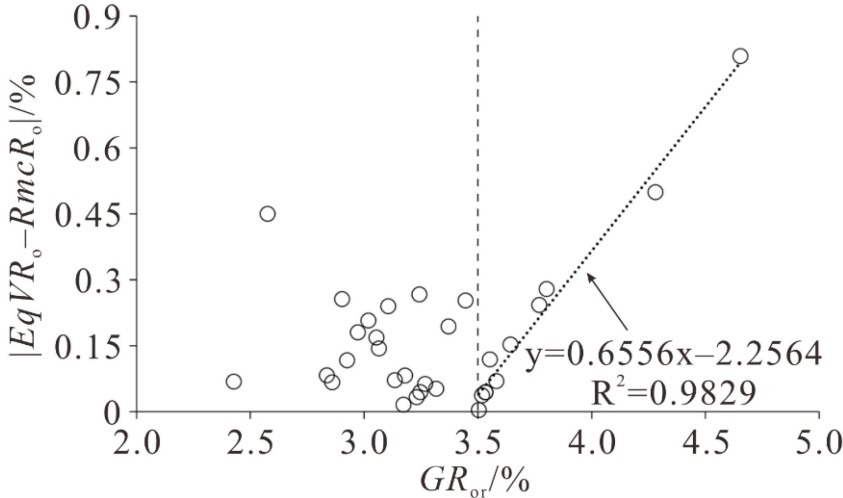

**Figure 12.** Analysis of the difference between $EqVR_o$ and $RmcR_o$.

Therefore, the relationship between $GR_{or}$ and $R_o$ during thermal evolution is not single linear. When $2.43\% < GR_{or} < 3.5\%$, there is a significant positive correlation between $RBS$ and $GR_{or}$ ($R^2 = 0.72$) (Figure 8c). The conversion equation of $RBS$ and $GR_{or}$ is obtained by regression of the two groups of parameters:

$$RBS = 17.639GR_{or} + 210.64 \tag{5}$$

When $GR_{or} > 3.5\%$, the trend of $RBS$ is reversed, so Equation (5) is no longer applicable at this stage. At this time, the law that $I_D/I_G$ increases rapidly with the increase in $GR_{or}$ is very obvious. The conversion equation of $I_D/I_G$ and $GR_{or}$ are obtained by regression of the two groups of parameters:

$$I_D/I_G = 0.2801GR_{or} - 0.3328 \tag{6}$$

Referring to Equations (3) and (4), the conversion equations of $EqVR_o$ and $GR_{or}$ can be obtained by using the media $RBS$ and $I_D/I_G$ as follows:

$$EqVR_o = 0.95GR_{or} + 0.1 \ (2.43\% < GR_{or} < 3.5\%) \tag{7}$$

$$EqVR_o = 0.33GR_{or} + 2.37 \ (GR_{or} > 3.5\%) \tag{8}$$

*5.2. Thermal Maturity of the Southern Sichuan Basin*

According to Equations (7) and (8), the $GR_{or}$ of the samples in this study are converted into $EqVR_o$ (Table 1, $EqVR_o$-5). The results show that the Wufeng–Longmaxi Formations in the southern Sichuan Basin are generally in the over-mature stage. Figure 13 shows that the Wufeng–Longmaxi Formations in the western Changning area have the highest maturity, the $EqVR_o$ ranging from 3.61% to 3.91%, with an average of 3.73%. Luzhou–western Chongqing followed, the $EqVR_o$ ranging from 2.92% to 3.57%, with an average of 3.30%. The $EqVR_o$ of eastern Changning ranges from 3.08% to 3.25%, with an average of 3.18%. The Weiyuan area has the lowest maturity, the $EqVR_o$ ranging from 2.41% to 3.12%, with an average of 2.80%.

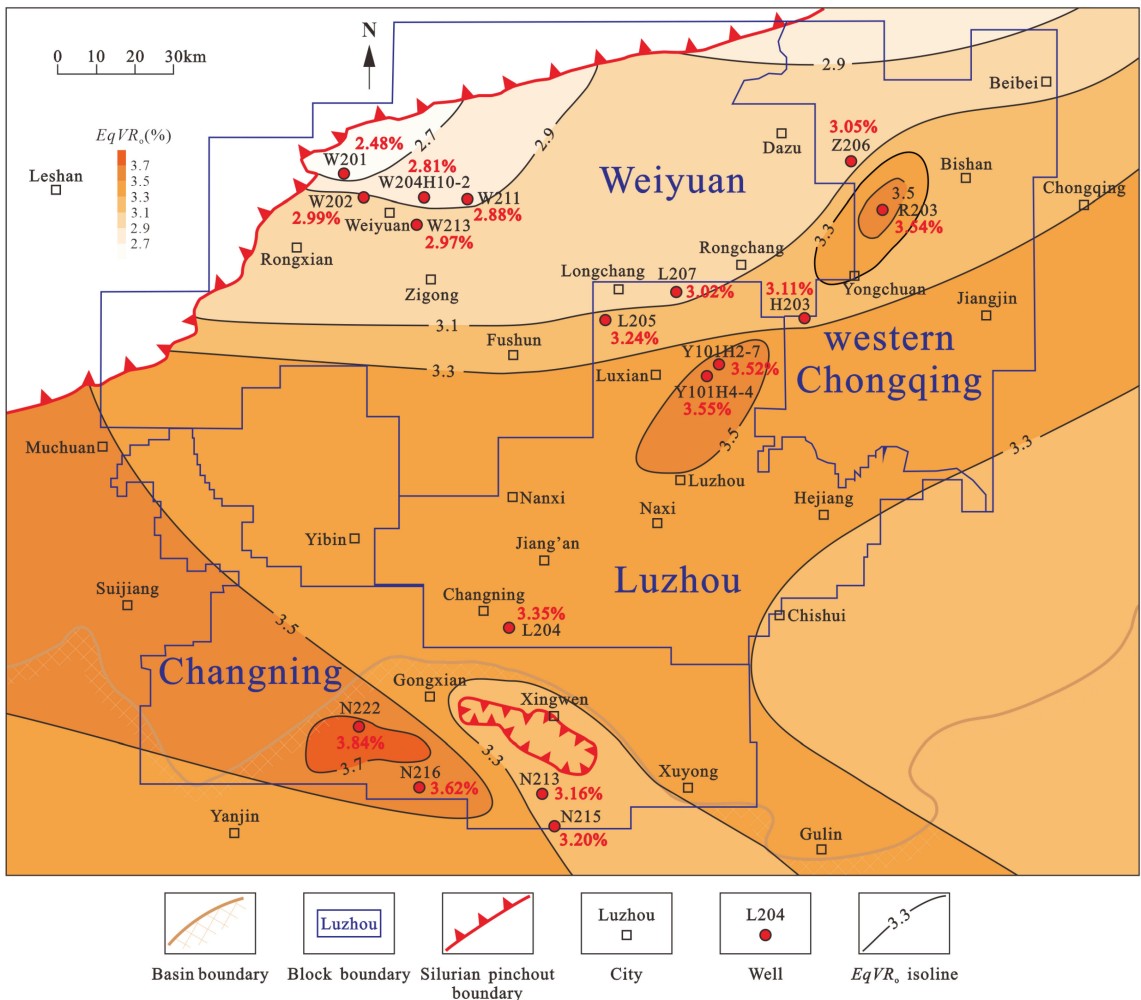

**Figure 13.** Equivalent vitrinite reflectance of the Wufeng–Longmaxi Shale in southern Sichuan Basin.

*5.3. Timing of Hydrocarbon Generation*

The thermal evolution and hydrocarbon generation history of the N222, Y101H4-4, N213 and W202 wells simulated by PetroMod V11 software (Figure 14) shows that the hydrocarbon generation processes of Wufeng–Longmaxi Formations in Weiyuan and Luzhou–western Chongqing are similar, while the hydrocarbon generation stage of the Changning area is significantly earlier than those of the other areas. The N222 and N213 wells experienced the first rapid maturation at the end of the Late Permian to the beginning of the Early Triassic, and rapidly reached the peak of oil generation. However, Y101H4-4 and W202 wells matured slowly and reached the peak of oil generation at the early stage of the Late Triassic. At this time, the N222 well entered the stage of thermal cracking and wet gas generation; the N213 well was also about to enter this stage. At the end of the Early Jurassic, the Y101H4-4 and W202 wells also entered the stage of wet gas generation. After the Middle Jurassic, the Wufeng–Longmaxi Formations in the southern Sichuan Basin reached the over-mature stage and began to generate dry gas. The dry gas generation stage lasted until the early stage of the Late Cretaceous.

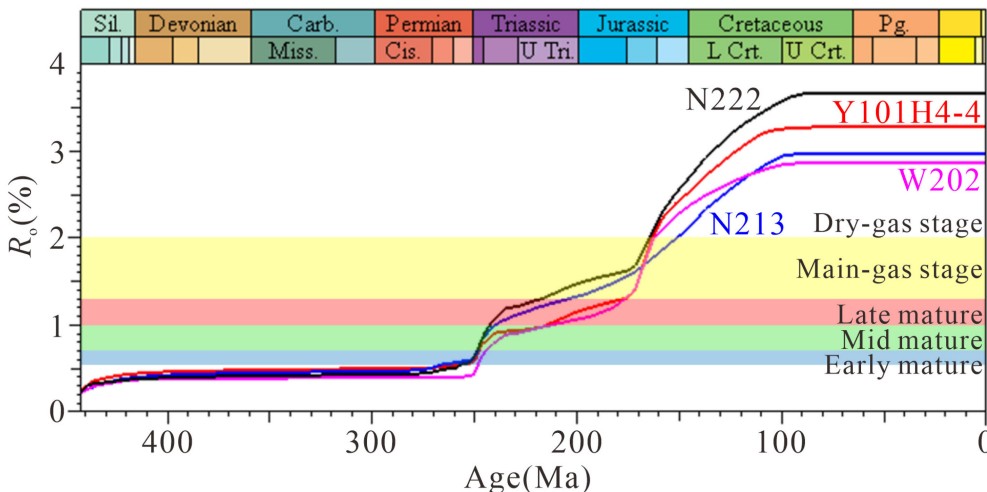

**Figure 14.** Thermal evolution and hydrocarbon generation history of the Wufeng–Longmaxi Formations in N222, Y101H4-4, N213 and W202 wells.

*5.4. Controlling Factors of Thermal Evolution Differences*

According to the plane distribution characteristics of the $EqVR_o$ of the Wufeng–Longmaxi Formations in the southern Sichuan Basin (Figure 13), the Weiyuan area near the Leshan–Longnüsi inherited paleo-uplift and eastern Changning area had the lower maturity, while Luzhou–western Chongqing located in the sedimentary center of the Longmaxi Formation and western Changning area near the Emeishan mantle plume had higher maturity. In particular, the N222 well in the western Changning area showed an abnormally high maturity phenomenon. The measured $GR_{or}$ of two samples from the $Long1_1^1$ bed are as high as 4.28% and 4.65%, and the $EqVR_o$ obtained by the conversion equation are 3.78% and 3.91%, respectively. Basin simulation results show that the maximum paleoburied depth of the N222, Y101H4-4 and W202 wells decreases successively. The maximum paleotemperature of the N222 and N216 wells in western Changning is significantly higher than those of other areas (Table 3). The differences in thermal evolution and hydrocarbon generation time sequence (Figure 14) indicate that the increasing rate of maturity of the Wufeng–Longmaxi Formations in the N222 well is obviously faster than that in other areas in the Late Permian. This may be related to the thermal effect of Emeishan basalt. After the Late Permian, the maturity of the Wufeng–Longmaxi Formations in the western Changning area was higher than that in other areas. The influence of the Emeishan mantle plume on the N213 well in the east of Changning is smaller than that in the west of Changning, so the maturity is lower. The Weiyuan area is close to the Leshan–Longnüsi inherited paleo-uplift, and its burial depth is lower than that of other areas in the southern Sichuan Basin, so its maturity is the lowest. Luzhou–western Chongqing is the sedimentary center of the shale of the Wufeng–Longmaxi Formations; it experienced a small magnitude of tectonic uplift in a late stage, so it has a larger burial depth. Therefore, its maturity is higher than those in the Weiyuan area and eastern Changning. Based on the above studies, it is inferred that the thermal evolution and oil and gas generation of the Wufeng–Longmaxi Shale in western Changning are controlled by the thermal effect of the Emeishan mantle plume and paleo-burial depth. The thermal evolution of eastern Changning, Weiyuan and Luzhou-western Chongqing is mainly controlled by paleo-burial depth.

## 6. Conclusions

The main organic matter in the Wufeng–Longmaxi Shale in the southern Sichuan Basin are graptolites and solid bitumen. Compared with solid bitumen, non-granular graptolites have a single source and obvious characteristics, and it is easy to measure the reflectance. In addition, the $GR_{or}$ of non-granular graptolites are more concentrated on the

vertical bedding sections. Thus, the $GR_{or}$ of non-granular graptolites are more suitable as an indicator of thermal maturity in Wufeng–Longmaxi Shale.

The correlation between Raman spectroscopic parameters of OM and $GR_{or}$ of non-granular graptolites indicate that the relationship between the $GR_{or}$ and the $R_o$ during thermal evolution is not single linear. Using the Raman spectroscopic parameters as intermediate variables, suitable conversion equations for the $GR_{or}$ and $EqVR_o$ for the Wufeng–Longmaxi Formations in the southern Sichuan Basin were established: $EqVR_o = 0.95GR_{or} + 0.1$ ($2.43\% < GR_{or} < 3.5\%$) and $EqVR_o = 0.33GR_{or} + 2.37$ ($GR_{or} > 3.5\%$).

The maturity of the Wufeng–Longmaxi Formations in the southern Sichuan Basin was more objectively and accurately evaluated by using the new equations. The results show that the western Changning area has the highest maturity, the $EqVR_o$ ranging from 3.61% to 3.91%, with an average of 3.73%. Luzhou–western Chongqing followed, the $EqVR_o$ ranging from 2.92% to 3.57%, with an average of 3.30%. The $EqVR_o$ of the eastern Changning area ranged from 3.08% to 3.25%, with an average of 3.18%. The Weiyuan area has the lowest maturity, the $EqVR_o$ ranging from 2.41% to 3.12%, with an average of 2.80%.

There are differences in thermal evolution and hydrocarbon generation history among the different areas in the southern Sichuan Basin. The thermal evolution of Wufeng–Longmaxi Shale in western Changning is controlled by the thermal effect of the Emeishan mantle plume and paleo-burial depth. However, the thermal evolution of eastern Changning, Weiyuan and Luzhou–western Chongqing is mainly controlled by paleo-burial depth.

**Author Contributions:** Conceptualization, J.M., R.P. and P.L.; methodology, J.M. and P.L.; formal analysis, P.L.; investigation, J.M. and P.L.; resources, J.M. and R.P.; writing—original draft preparation, P.L. and J.M.; writing—review and editing, J.M. and X.Y.; visualization, P.L., T.Y. and N.Z.; supervision, R.P. and J.M.; project administration, J.M. and R.P.; funding acquisition, R.P. and J.M. All authors have read and agreed to the published version of the manuscript.

**Funding:** This research was funded by the Scientific and Technological Innovation Team of the outstanding Young and Mid-age in Colleges of Hubei Province (T201905) and the Young Scientists Fund of the National Natural Science Foundation of China (41402114).

**Institutional Review Board Statement:** Not applicable.

**Informed Consent Statement:** Not applicable.

**Data Availability Statement:** Data will be made available upon request.

**Acknowledgments:** The authors would like to thank Yiqing Zhu from the Shale Gas Research Institute of PetroChina Southwest Oil and Gasfield Company, Chengdu, Sichuan, China, for his support and scientific guidance. We appreciate the valuable comments from the editors and anonymous reviewers.

**Conflicts of Interest:** The authors declare no conflict of interest.

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
