# Peer review of "Characteristics and Differences Analysis for Thermal Evolution of Wufeng–Longmaxi Shale, Southern Sichuan Basin, SW China"

_minerals, doi:10.3390/min12070906_

Round 1

Reviewer 1 Report

This study has done an excellent experiment job in evaluating the reflectivity of graptolites and solid bitumen. It has certain reference value for the maturity and thermal evolution history of marine shale in the southern Sichuan Basin. There are several questions that need to be discussed with the author.

1. The authors divide graptolites into granular graptolites and non-granular graptolites. However, they have completely different characteristics. In the study by Henrik I. Petersen (2013), a vitrinite-like particle was described. I don't know, whether the granular graptolites mentioned by the author are vitrinite-like particles?

2. The characteristic parameters of Raman spectra change regularly with the increase of graptolite reflectivity, but usually turn around at 3.5%. What is the reason for this?

3. Some pictures are not clear, such as Figures 4, 8, etc.

4. In the figure 12, is the contour plot drawn from only 33 samples?

5. Please check carefully to avoid writing errors.

Author Response

Response to Reviewer 1 Comments

Point 1: The authors divide graptolites into granular graptolites and non-granular graptolites. However, they have completely different characteristics. In the study by Henrik I. Petersen (2013), a vitrinite-like particle was described. I don't know, whether the granular graptolites mentioned by the author are vitrinite-like particles?

Response 1: The granular graptolites mentioned in this paper are not vitrinite-like particles. In the study by Petersen et al. (2013), the vitrinite-like particles do not possess the diagnostic features of fragments from graptolites and exhibit a more varied morphology. The outline of vitrinite-like particles is irregular mosaic structure, and their distribution direction is also irregular. However, the granular graptolites in this paper have a clear outline, usually oval, and its long axis direction is often consistent with the bedding direction. Due to the reflectance of graptolites and vitrinite-like particles in the same sample was identical, Petersen et al. (2013) thought that the vitrinite-like particles were fragments of graptolites. However, animal organisms have specific morphology. From the irregular outline of vitrinite-like particles described by Petersen et al. (2013), they are more likely to be the pyrolysis residues of algae. The granular graptolites in this paper have a specific shape, so we think they are fragments of graptolites.

Point 2: The characteristic parameters of Raman spectra change regularly with the increase of graptolite reflectivity, but usually turn around at 3.5%. What is the reason for this?

Response 2: According to the size and structure of aromatic compounds in organic matter, the evolution of Raman band dispersion during coalification and graphitization could be divided into two stages. The first stage showed a downshift of D band, which was assumed to be related to the progressive growth of linear aromatic structures. The second stage was characterized by diverging D band positions, which is caused by the growth of compact polycyclic aromatic hydrocarbons. The graptolite random reflectance corresponding to the boundary of these two stages is about 3.5%. In contrast, G peak displayed a small trend of higher wavenumber shift at the beginning, and then stabilized at around 1600 cm-1. The band separation gradually increased under the control of the peak position. However, the band separation no longer increased but decreased when the graptolite random reflectance reached about 3.5%.

Point 3: Some pictures are not clear, such as Figures 4, 8, etc.

Response 3: Thank you for the suggestion. We have redrawn or modified Figure 1, 2, 3, 4, 5, 8, 9, 12 and 13.

Point 4: In the figure 12, is the contour plot drawn from only 33 samples?

Response 4: Using 33 new samples from different parts of southern Sichuan Basin, we systematically analyzed the relationship between graptolite random reflectance, solid bitumen reflectance, Raman spectral parameters of kerogens and equivalent vitrinite reflectance. Then, we calibrated the maturity of Wufeng–Longmaxi Formations in southern Sichuan more accurately. On the basis of previous studies on the maturity of Wufeng–Longmaxi Formations in Sichuan Basin, we drew Figure 12 (now Figure 13) with more accurate data. Our equivalent vitrinite reflectance isoline map referred to the maturity trend of Wufeng–Longmaxi Formations in Sichuan Basin from previous studies.

Point 5: Please check carefully to avoid writing errors.

Response 5: Thank you for the suggestion. We have carefully proofread and corrected the language problems and writing errors in the article. In order to improve our article to a level suitable for publication, the manuscript has undergone English language editing by MDPI.

The detailed modification have been included in the revised manuscript file.

Reviewer 2 Report

Dear Authors,

Below you find my main objections to the article, while the detailed corrections have been included in the manuscript file.

1. Title

The title should be changed. I propose a new one "Thermal maturity of the lower Paleozoic sediments determined by Raman spectroscopy, Southern Sichuan Basin, SW China"

2. Abstract

The abstract must be re-edited. For me, it is a conglomerate of various thoughts and the reader does not know what is really important

Try to include in your abstract the following information: what was done and why?  how did you do that? Give Results: briefly; and Conclusion: as summary

3. Introduction

the introduction is an important part of the manuscript because it introduces the reader to the subject and the whole idea of the manuscript. after reading Your introduction, I am confused because I do not know what new your research brings to the recognition and description of the Paleozoic shales thermal maturity.

Why Your research is significant? Underline this aspect

4. Geological setting

The location map is unclear. You should present the geological setting from general to detail. The readers should know the relations between the study area and China territory.

Geology of the area study is absolutely mandatory. Include the sedimentological log of the shales and describe the main evolution stages of the Sichuan basin.

5. Samples and methods

It is unclear how many samples were collected from each well. Which samples were used for each method? why such samples were selected? Samples description must be clear and detailed about the selection criteria.

Why did you choose such methods? what are their strengths?

Table 1 must be improved and corrected.

6. Results

check the definition of the „maceral” in Stach’s textbook of coal petrology 1975.

In high matured samples with anisotropy evidence, GRor min and max should be measured. You do not present such results

The figures are not complete. You show a few images or plots with no description

Level of English is Your manuscript is poor. I did not a detailed correction of that but reading the text is really confused. There are many mental abbreviations in each paragraph. Many times I lost the main idea of the chapter and had to return to read it again.

Author Response

Response to Reviewer 2 Comments

Point 1: Title

The title should be changed. I propose a new one "Thermal maturity of the lower Paleozoic sediments determined by Raman spectroscopy, Southern Sichuan Basin, SW China".

Response 1: In this paper, we systematically analyzed the relationship between graptolite random reflectance, solid bitumen reflectance, Raman spectral parameters of kerogens. By using Raman spectral parameters as mediators, the conversion equations between graptolite random reflectance and equivalent vitrinite reflectance of Wufeng–Longmaxi Formations in the southern Sichuan Basin were established. Raman spectroscopy is just one of the methods. In the second half of the article, we reconstruct the burial history, thermal evolution history and hydrocarbon generation history of Wufeng–Longmaxi Shale in different areas of southern Sichuan Basin through basin modelling, and expound the factors for the differences of thermal maturity. Therefore, we think the original title is more comprehensive and accurate.

Point 2: Abstract

The abstract must be re-edited. For me, it is a conglomerate of various thoughts and the reader does not know what is really important. Try to include in your abstract the following information: what was done and why? how did you do that? Give Results: briefly; and Conclusion: as summary.

Response 2: Thank you for the suggestion. We have modified the abstract according to the comment.

Point 3: Introduction

The introduction is an important part of the manuscript because it introduces the reader to the subject and the whole idea of the manuscript. After reading your introduction, I am confused because I do not know what new your research brings to the recognition and description of the Paleozoic shales thermal maturity. Why Your research is significant? Underline this aspect.

Response 3: Thank you for the suggestion. We have modified and supplemented the introduction according to the comment. In fact, in previous researches, there are limitations to using a single indicator to study the thermal maturity of the Lower Paleozoic sediments. Our research presents a systematic analysis of the reflectance of graptolites and solid bitumen, as well as Raman spectroscopy characteristics of kerogens. Through this way, we make a more objective and accurate evaluation of the thermal maturity of the Wufeng–Longmaxi Formations in southern Sichuan Basin. And we also analyzed the factors for the differences in the maturity of Wufeng–Longmaxi Formations in different areas of southern Sichuan Basin, which were rarely involved in previous studies.

Point 4: Geological setting

The location map is unclear. You should present the geological setting from general to detail. The readers should know the relations between the study area and China territory. Geology of the area study is absolutely mandatory. Include the sedimentological log of the shales and describe the main evolution stages of the Sichuan basin.

Response 4: Thank you for the suggestion. We have redrawn Figure 1 and supplemented the geological settings according to the comment.

Point 5: Samples and methods

It is unclear how many samples were collected from each well. Which samples were used for each method? why such samples were selected? Samples description must be clear and detailed about the selection criteria. Why did you choose such methods? what are their strengths? Table 1 must be improved and corrected.

Response 5: Thank you for the suggestion. We have modified Chapter 3 and corrected Table 1 according to the comment. In fact, in the early stage of the study, we took a sample every 0.5–1 m according to the thickness of Wufeng Formation to Long11 sub-member from each well. Combined with the results of TOC, XRD, major and trace elements test, Ar-ion polishing FE-SEM analysis, rock thin section observation and N2 and CO2 adsorption experiments, we selected 1–4 samples from each well. The lithology of the samples is mainly black siliceous shale. Each sample was used for each test. These methods were chosen based on the experience of previous researchers. The comprehensive use of these methods in our research can make up for their limitations.

Point 6: Results

Check the definition of the „maceral” in Stach’s textbook of coal petrology 1975. In high matured samples with anisotropy evidence, GRor min and max should be measured. You do not present such results. The figures are not complete. You show a few images or plots with no description.

Response 6: We have checked the definition of the “maceral” in Stach’s textbook of coal petrology 1975 and modified this expression throughout the text. We have also added descriptions to some images, such as Figure 1, 2, 3 and 5. Graptolites in high–over-mature shale have obvious anisotropy, graptolite maximum reflectance (GRomax) and graptolite minimum reflectance (GRomin) can be observed in sections parallel and normal to bedding, respectively. However, due to an unusual burial process or sample preparation process, it is difficult for graptolites in these samples to be truly parallel to bedding or normal to bedding. Recently, Luo et al. (2018, 2019, 2020) found that there is a significant positive correlation between graptolite random reflectance (GRor) and GRomax on sections parallel to bedding. The GRor is a better thermal maturity indicator and is more precise due to the more convenient testing process and smaller standard deviation in comparison with the GRomax. In over-mature stage, the coalification trend for graptolites based on the GRomax and GRomin is different from that for heat-affected coal (vitrinite), coal-graphite, and bitumen. In addition, the anisotropy of graptolite is not the focus of this paper. Thus, we only observed the anisotropy of graptolites in samples, but did not measure the GRomax and GRomin.

Point 7: Level of English of your manuscript is poor. I did not a detailed correction of that but reading the text is really confused. There are many mental abbreviations in each paragraph. Many times I lost the main idea of the chapter and had to return to read it again.

Response 7: Thank you for the suggestion. We have carefully proofread and corrected the language problems and writing errors in the article. We have modified or canceled some abbreviations in the text to avoid confusion. In order to improve our article to a level suitable for publication, the manuscript has undergone English language editing by MDPI.

The detailed modification have been included in the revised manuscript file.

Reviewer 3 Report

More detailed information on graptolites and bitumen reflectance has to be  added to the introduction, references are in the literature, but the text does not correspond

There are not mentioned publications dealing with the graptolite reflectance in the same basin - Application and Research on Macroscopic Identification of Bitumen and Graptolite in Shale and Reflectance Detection ( Open Journal of Yangtze Gas and Oil, 2018, 3, 11-20)

I miss sensitivity analyses for your burial model, majority of conclusions are based on not very clearly presented calibration, especially if you have just one 1D model.  I have big doubts about the presented 1D model, the sensitivity analysis is not presented, this is necessary especially when the model is only from one well, I suggest to provide models of other wells from which the authors have data.

the results of graptolite reflectance are partially evaluated, it is necessary to add to the introduction already known facts about the development of graptolite reflectance with increasing thermal maturity and then improve the interpretation of results

Add a lithostratigraphic column

Add map of broder area – as a minimum a position within China,

r. 111 - The total organic carbon is mainly in the range of 2%~5% - what is meant by this - is not appropriate to describe your results in this way in a professional publication

r. 340-342 – It is not specific what drilling data is meant

Chapter 4.3.1 belong to introduction section not to results

Chapter 4.3.2 Å™. 374-380 its not possible to conclude without sensitivity analyses,

Fig 13 - calibration is very unreliable

Author Response

Response to Reviewer 3 Comments

Point 1: More detailed information on graptolites and bitumen reflectance has to be added to the introduction, references are in the literature, but the text does not correspond.

Response 1: Thank you for the suggestion. We have added more information about the reflectance of graptolites and solid bitumen to the introduction. We have also added more references.

Point 2: There are not mentioned publications dealing with the graptolite reflectance in the same basin – Application and Research on Macroscopic Identification of Bitumen and Graptolite in Shale and Reflectance Detection (Open Journal of Yangtze Gas and Oil, 2018, 3, 11-20).

Response 2: Thank you for the suggestion. We have carefully read this important article and added it into the introduction according to the comment. This article is very helpful for us to revise our manuscript. In particular, the maturity evaluation of the Wufeng–Longmaxi Formations in Northwest Guizhou is helpful for us to have a more comprehensive understanding of the maturity of the Lower Paleozoic in South China.

Point 3: I miss sensitivity analyses for your burial model, majority of conclusions are based on not very clearly presented calibration, especially if you have just one 1D model. I have big doubts about the presented 1D model, the sensitivity analysis is not presented, this is necessary especially when the model is only from one well, I suggest to provide models of other wells from which the authors have data.

Response 3: In the new Chapter 4.3.3, we used the equivalent vitrinite reflectance and the calculated maximum paleo-temperature of the Wufeng–Longmaxi Formations to calibrate the model. It can be seen that the simulation results are in good agreement with the measured results. We also compared our simulation results with the previous studies on the burial history and thermal evolution history of Wufeng–Longmaxi Formations in different areas of southern Sichuan Basin. The results show that the maturity evolution curve and hydrocarbon generation curve simulated by our model are consistent with the previous research results. The above results show that our simulation results are reliable. In fact, a total of seven wells have been used to simulate the burial history and thermal evolution history. We found that the simulation results of different wells from the same area just have little difference. Thus, in this paper, we showed the simulation results of four wells in different areas of southern Sichuan Basin. These four wells have the most complete data, and the simulation results should be the most accurate.

Point 4: The results of graptolite reflectance are partially evaluated, it is necessary to add to the introduction already known facts about the development of graptolite reflectance with increasing thermal maturity and then improve the interpretation of results.

Response 4: Thank you for the suggestion. We have added more information about the development of graptolite reflectance and solid bitumen reflectance with increasing thermal maturity to the introduction. We have also modified the relevant results according to the comment.

Point 5: Add a lithostratigraphic column.

Response 5: We have added a lithostratigraphic column in Figure 1.

Point 6: Add map of border area – as a minimum a position within China.

Response 6: We have added the location map of Sichuan Basin in China in Figure 1.

Point 7: r. 111 – The total organic carbon is mainly in the range of 2%~5% – what is meant by this – is not appropriate to describe your results in this way in a professional publication.

Response 7: Thank you for the suggestion. The precedent version of the expression has been modified, becoming “According to statistical data, TOC values of Wufeng–Longmaxi Shale range from 0.4% to 18.4%, with an average of 2.5%” and “The main research sections are the upper Wufeng Formation and Long111-Long114 beds, with a thickness of 40–130 m and TOC value over 2%”.

Point 8: r. 340–342 – It is not specific what drilling data is meant.

Response 8: We have added the specific meaning of drilling data in the article according to your comment. The drilling data used in our research are mainly strata division data, including the name of stratigraphic units and their thickness.

Point 9: Chapter 4.3.1 belong to introduction section not to results.

Response 9: In this section, we described the estimated erosion amount and heat flow in different areas of southern Sichuan Basin. Due to the lack of relevant research in southern Sichuan Basin, we determined the erosion amount and heat flow in different areas in southern Sichuan Basin by referring to the previous research results in other areas of the Sichuan Basin, combined with the structural evolution and actual stratigraphic data in southern Sichuan. According to the article structure and data sources, we think it is more appropriate to keep this chapter in its current position.

Point 10: Chapter 4.3.2 r. 374–380 its not possible to conclude without sensitivity analyses. Figure 13 – calibration is very unreliable.

Response 10: We have responded to the comment of sensitivity analyses in Response 3.

The detailed modification have been included in the revised manuscript file.

Round 2

Reviewer 2 Report

Dear Authors,

I have found your manuscript improved significantly. I appreciate you corrected actually my comments. I attached the manuscript with my few detailed remarks.

Best regards,

The Reviewer

Author Response

Dear Reviewer,

Thank you very much for your valuable suggestions. Under your guidance, we have further revised the manuscript.

Best regards,

The Authors

Reviewer 3 Report

For the next research I suggest to do 2D modelling in case you have just limited depth interval for borehole. 

Author Response

Dear Reviewer,

Thank you very much for your valuable suggestions on our manuscript and research. In future research, we will try to use 2D modelling to improve the reliability of our research.

Best regards,

The Authors